# SIgA structures bound to *Streptococcus pyogenes* M4 and human CD89 provide insights into host-pathogen interactions

Qianqiao Liu ®[1] & Beth M. Stadtmueller ®[1,2,3] ✉

Immunoglobulin (Ig) A functions as monomeric IgA in the serum and Secretory (S) IgA in mucosal secretions. Host IgA Fc receptors (FcαRs), including human FcαR1/CD89, mediate IgA effector functions; however, human pathogen *Streptococcus pyogenes* has evolved surface-protein virulence factors, including M4, that also engage the CD89-binding site on IgA. Despite human mucosa serving as a reservoir for pathogens, SIgA interactions with CD89 and M4 remain poorly understood. Here we report cryo-EM structures of M4-SIgA and CD89-SIgA complexes, which unexpectedly reveal different SIgA-binding stoichiometry for M4 and CD89. Structural data, supporting experiments, and modeling indicate that copies of SIgA bound to *S. pyogenes* M4 will adopt similar orientations on the bacterium surface and leave one host FcαR binding site open. Results suggest unappreciated functional consequences associated with SIgA binding to host and bacterial FcαRs relevant to understanding host-microbe co-evolution, IgA effector functions and improving the outcomes of group A *Streptococcus* infection.

Immunoglobulin (Ig) A is an antibody class expressed in mammals, where it functions in monomeric (m) and secretory (S) forms; the mIgA is the predominant form in serum whereas SIgA is the most abundant antibody present in mucosal secretions, where it protects the host epithelium against microbes and toxins[1]. Both forms of IgA contain fragment antigen-binding (Fab) and fragment crystallizable (Fc) regions. The Fab is responsible for binding to antigens, whereas the Fc region interacts with other components of the immune system, including IgA Fc Receptors (FcαRs) such as CD89 (a.k.a. FcαR1), that are expressed on the surface of some immune system cells and can promote downstream effector functions such as phagocytosis and respiratory burst[2].

CD89-initiated effector functions are thought to occur when multiple copies of the receptor bind to multiple IgA on IgA-antigen complexes, promoting clustering and crosslinking of the receptors and associated cell signaling through FcR γ-chain complexes[3]. In turn, binding to free IgA does not promote crosslinking and has been proposed to transduce anti-inflammatory signals instead[4]. CD89 effector

functions have been studied primarily in the context of mIgA, consistent with the observation that cells expressing CD89 are more prevalent in the serum than mucosa. However, phagocyte populations also sample and/or patrol mucosal surfaces, and SIgA has been associated with CD89-expressing polymorphonuclear leukocytes (PMN) activity in the presence of accessory molecule Mac-1 (CD11b/CD18)[5]. Whereas the binding of host FcαRs is likely to elicit an effector immune response, IgA can also be bound by bacterial proteins, for example, M4 (a.k.a. Arp4) and M22 (a.k.a Sir22) from *Streptococcus pyogenes*, the β-protein from group B streptococci (GBS), and SSL7 from *Staphylococcus aureus*, interactions that may allow pathogenic bacteria to evade host IgA responses[6–10]. Despite significance, the functional consequences of IgA interactions with host FcαRs and bacterial proteins, sometimes referred to as decoys, remain poorly understood.

The monomeric and secretory forms of IgA that mediate host interactions with antigens, FcαRs, and bacterial proteins adopt at least two distinct molecular structures. The mIgA contains two IgA heavy chains and two light chains, resulting in two Fabs and one Fcα. The Fcα

[1]Department of Biochemistry, University of Illinois Urbana-Champaign, Urbana, Illinois 61801, USA. [2]Department of Biomedical and Translational Sciences, Carle Illinois College of Medicine, University of Illinois Urbana-Champaign, Urbana, Illinois 61801, USA. [3]Carl R. Woese Institute for Genomic Biology, University of Illinois, Urbana, Illinois 61801, USA. ✉e-mail: bethms@illinois.edu

contains two copies of immunoglobulin constant domains, Cα2 and Cα3, and adopts a rigid structure linked to two Fabs. Fcα contains two FcαR binding sites, one located at each Cα2 and Cα3 interface (Fig. 1a). The Cα2/Cα3 interdomain has been recognized as a "hot spot" for host FcαRs and bacterial IgA-binding proteins[6,11,12]. SIgA is polymeric and is typically composed of two mIgA that assemble with one joining-chain (JC) to form dimeric (d) IgA, which is bound by the polymeric Ig-receptor (pIgR) ectodomain, called secretory component (SC). The structures of dIgA and SIgA reported in 2020, revealed two Fcα connected through four C-terminal motifs called tailpieces (Tps) and one JC, which contacts each heavy chain uniquely[13–15]. This results in two structurally unique Fcαs that can be designated as Fc$_{AB}$ (Fcα containing heavy chains A and B) and Fc$_{CD}$ (Fcα containing heavy chains C and D). Together, Fc$_{AB}$ and Fc$_{CD}$ adopt an asymmetric, bent and tilted conformation with distinct concave and convex sides. In SIgA, the SC, which comprises five Ig-like domains (D1-D5), is bound toward one side and protrudes from the center of the molecule contributing further asymmetry. Overall, the SIgA conformation is predicted to constrain the possible positions that Fabs can adopt and leave two FcαR binding sites accessible (Fig. 1b). Notably, these two sites are located on the convex side, one on Fc$_{AB}$ and one on Fc$_{CD}$, whereas sites on the concave side are occluded by the JC and the SC. Thus, SIgA is thought to have the same number of accessible FcαR binding sites as mIgA but instead of two sites on one Fcα that are horizontally oriented relative to each other, it has one site on two Fcαs that are vertically oriented relative to each other (Fig. 1)[13]. The functional significance of these differences remains a topic of investigation; however, it is notable that two receptor binding sites have been evolutionarily preserved in both mIgA and SIgA.

Both mIgA and SIgA can be bound by group A streptococcus (GAS; *Streptococcus pyogenes*)[16]. GAS has long been considered a significant human pathogen that continues to be one of the top ten causes of mortality worldwide from infectious diseases[17]. The clinical manifestations of this Gram-positive bacterium are wide-ranging and diverse, encompassing both mild and common local infections, such as tonsillitis (a.k.a. Strep throat) and impetigo, as well as life-threatening systemic invasions like toxic shock and sepsis[18,19]. While

non-invasive disease has ~1000-fold higher incidence than invasive disease, approximately 1.78 million new invasive infections are reported each year and are associated with over 160,000 deaths[17,20]. GAS asymptomatic carriers are also common, with a prevalence of 12% among healthy children[21]. Typically, invasive infections encounter mIgA in the serum, whereas non-invasive, region infections encounter SIgA originating from mucosal secretions (e.g., in the upper respiratory tract) as well as mIgA that perfuses inflamed tissues. Exceptions to this include GAS skin infections such as impetigo where likely, only mIgA would be encountered, and (asymptomatic) GAS colonization of mucosal secretions where inflammation is limited and SIgA is expected to dominate.

The most extensively studied GAS virulence factors are a family of serotype-specific proteins called M proteins, which include IgA-binding proteins M4, M11, M22, M28, M48, M60, and M85[22,23]. M proteins adopt a coiled-coil structure that protrudes up to 60 nm from the bacterium cell and can bind a variety of host extracellular matrix and serum proteins[24,25]. Variability in the M protein ectodomain is correlated with the ability to bind specific host factors; for example, some M proteins bind IgG, some IgA, some fibrinogen, and some other host factors[26,27]. These interactions are thought to contribute to the bacterium's virulence primarily by modulating the host immune response (e.g., limiting phagocytosis).

M4 and M22 are well-characterized M proteins found in GAS strains M4 and M22, respectively, both of which can bind IgA and have widespread global distribution; M4 ranked in the top five most common serotypes identified in a systemic review of global GAS distribution[8,9,28,29]. Mutational analysis of both M4 and M22 revealed a 29-residue, minimal IgA-binding region located within the ectodomain with similar, but not identical, sequences[9] (Fig. 1c). M22 also binds IgG at a site overlapping with the IgA-binding site[8]. Despite significance, how M proteins interact with host ligands remains unclear. Existing structural data includes crystal structures of an M1 fragment containing the HVR and a repetitive sequence element within the ectodomain and the HVR of M2, M22, M28, and M49, none of which detail the structure of the IgA-binding region or IgA-M protein interactions[30,31]. Published analysis indicated that M4 and M22 binding to IgA is

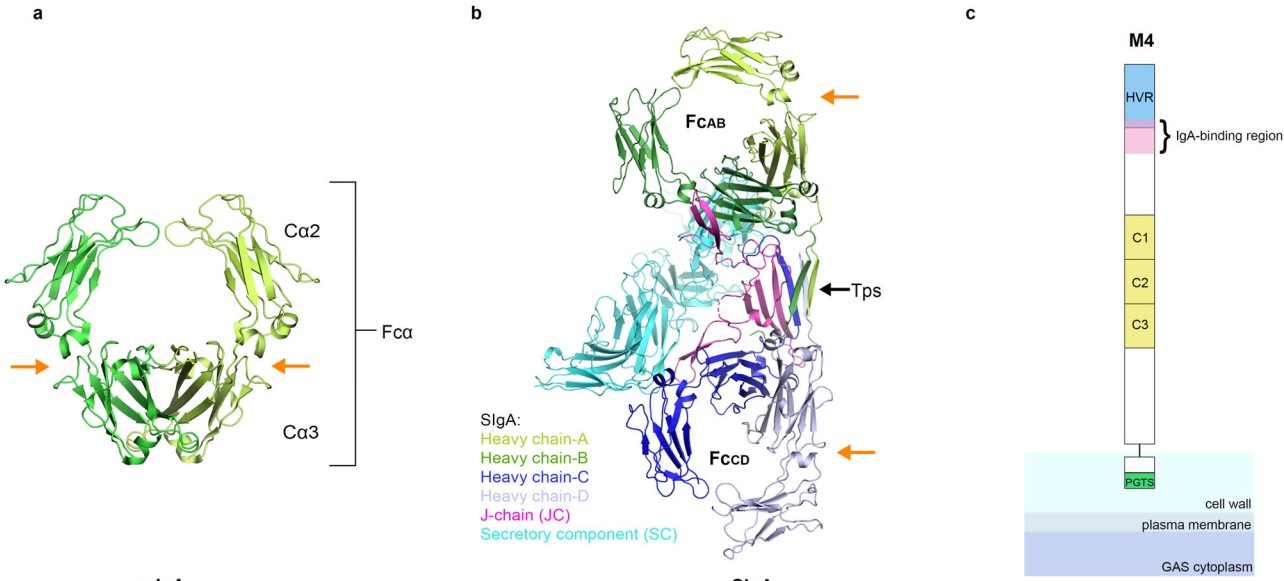

**Fig. 1 | Structures of mIgA, SIgA, and schematic of M4. a** The structure of mIgA lacking Fabs (Fcα; PDB code 1OW0 with CD89 removed). **b** The structure of human SIgA lacking Fabs (PDB code 6UE7). Structures are colored according to the key; one Cα2 and one Cα3 domain are labeled along with Fc$_{AB}$ and Fc$_{CD}$. The SIgA tailpieces (Tps) are labeled and indicated with a black arrow and the FcαR binding sites are indicated by orange arrows. **c** Schematic of M4 depicted attached to the GAS cell wall through a C-terminal proline-glycine-threonine-serine (PGTS)-rich domain and a cell-wall-associated domain, which are followed by an ectodomain that includes three C-repeats, an IgA-binding region and an N-terminal hypervariable domain (HVR).

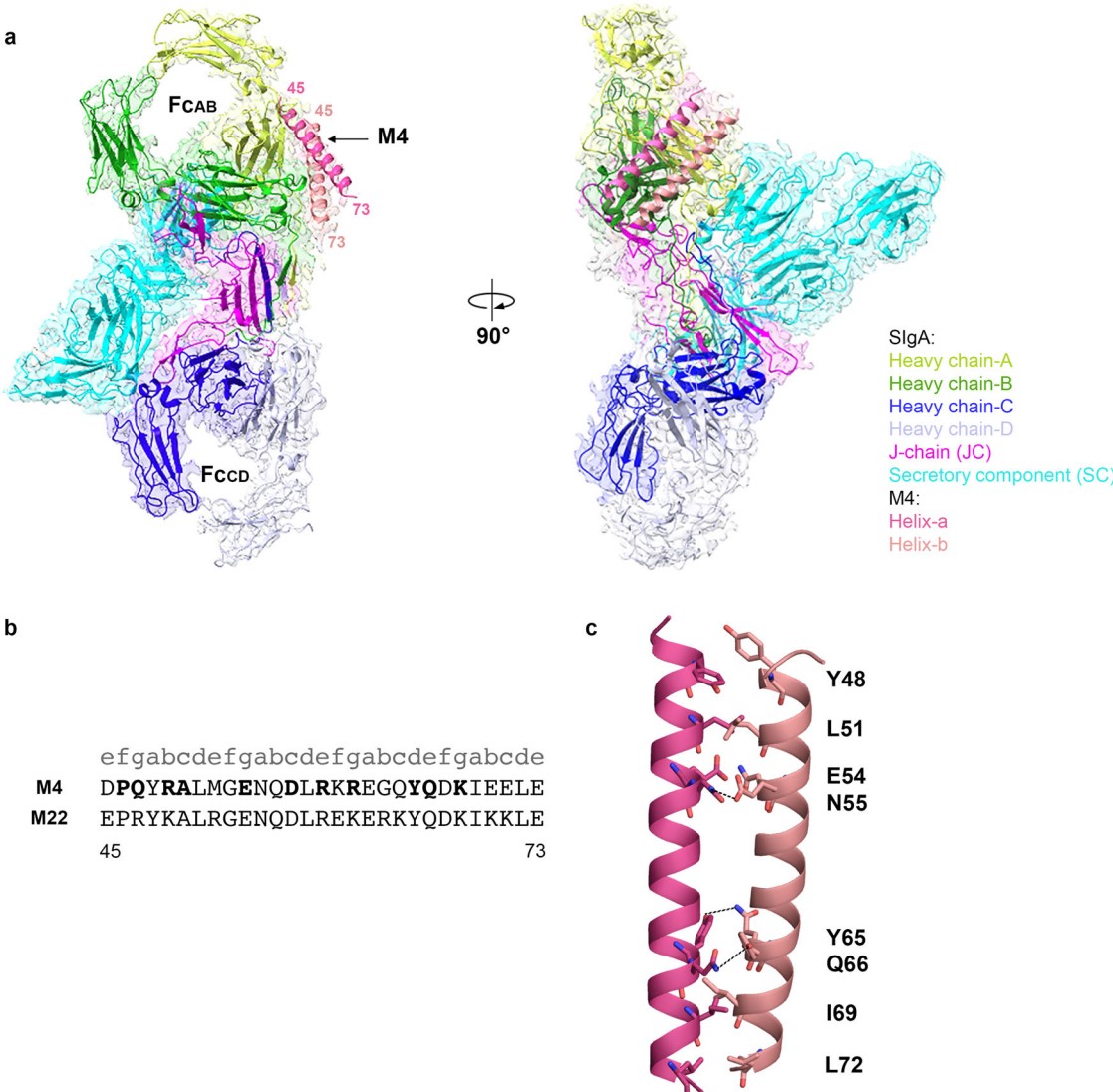

**Fig. 2 | SIgA-M4 Structure. a** The Cryo-EM structure of the M4 fragment (residues 45–73) in complex with human SIgA is shown as a cartoon with a partially transparent map in two orientations. The model is colored according to the key, M4 is labeled, and Fabs are disordered. **b** Sequence alignment of the IgA-binding regions of M4 and M22 with their heptad registrations shown above; numbering corresponds to both sequences, and M4 residues interacting with IgA are depicted in bold. **c** Cartoon representation showing interchain interactions between M4 helix a and b. Polar interactions are shown as dotted lines.

dependent on residues located at the Cα2/Cα3 interface, including residues bound by host CD89 and *S. aureus* SSL7[2,6,7]. The overlapping interface of M4 and CD89 suggested that M4 may interfere with IgA effector functions, and studies suggested that M4 protein can block respiratory burst in neutrophils in vitro[11]. However, the role of M4 (and CD89) in disease pathogenesis remains poorly understood, and published studies have focused on mIgA despite mucosal routes of entry.

To investigate how M proteins interact with SIgA and how those interactions compare to host FcαRs, we determined two Cryo-EM structures, one of SIgA in complex with M4 and one of SIgA in complex with CD89. Together structures indicate that M4 and CD89 share only a subset of IgA interface residues and bind through different angles of approach. This leads to differences in binding stoichiometry, with M4 occupying only one FcαR binding site on SIgA while leaving the other site sterically accessible.

## Results

### Structure determination of the SIgA-M4 complex

In order to better understand how bacterial virulence factors such as M4 engage SIgA, we designed an M4 expression construct encoding

the ectodomain (residues 1–314) and combined the resulting protein with purified, recombinant human SIgA1 to create an SIgA-M4 complex. Using this complex, we determined the SIgA-M4 cryo-EM structure to an average resolution of 3.1 Å (FSC = 0.143) (Fig. 2a). Map quality and resolution were sufficient to refine the positions of main chain and side chain atoms for the majority of residues in the complex with the exception of Fab residues, which were disordered (Supplementary Fig. 1). The final structure included a single M4 dimer bound to one SIgA; M4 residues 45–73, which include the entire known IgA-binding domain, were built whereas the positions of residues more distal from the binding site were poorly resolved or disordered, likely due to the inherent flexibility of the M4 dimer's extended, coiled-coil structure. SIgA is superimposable with a previously published human SIgA1 structure with an RMSD of 0.708[14].

The M4 adopts an α-helical, parallel coiled-coil homodimer as observed for other M proteins[26]. The M4 fragment termini are stabilized by hydrophobic interactions, including two ideal heptads, characterized by having Tyr, Ile, or Leu at both *a* and *d* positions[32]. Hydrophobic residues at the *a* and *d* positions (Y48, L51, I69, L72) are packed against each other and stabilize the coiled-coil (Fig. 2b, c). We

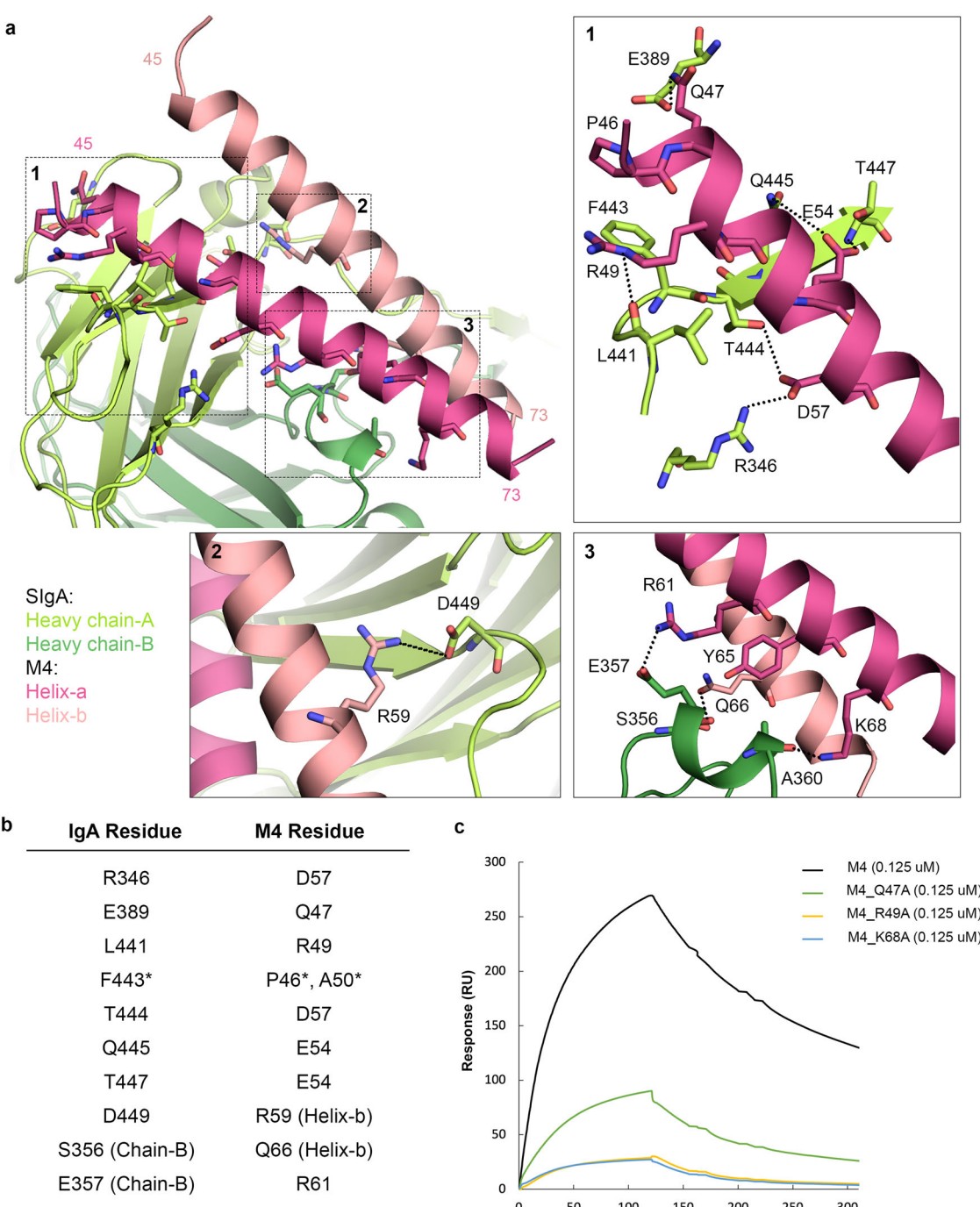

**Fig. 3 | The M4-SIgA interface. a** Structure of the M4-SIgA interface. The boxed regions are enlarged (1–3) and depict the (1) M4 helix-a interaction with HC-A, (2) M4 helix-b interactions with HC-A, and (3) M4 interactions with HC-B. Black dotted lines indicate potential hydrogen bonds. **b** Polar and hydrophobic interactions between IgA and M4. Asterisks indicate residues contributing to hydrophobic interactions. **c** M4 binding analysis by surface plasmon resonance (SPR). The sensorgram shows the response of M4 (black) and individual M4 point mutants Q47A (green), R49A (yellow), and K68A (blue) binding to immobilized human SIgA1. All analyte concentrations shown are 0.125 uM. Sensorgrams, including a complete concentration series for each analyte, binding models and rate constants, can be found in Supplementary Fig. 3 and Supplementary Table 2. Source data are provided in the Source Data file.

also observed polar interactions stabilizing the center of the M4 fragment, including E54 which forms a hydrogen bond with N55, and Y65 which forms a hydrogen bond with Q66 (Fig. 2c). A subset of residues between P46-K68 in both helices contact SIgA, with one (helix-a) forming the majority of contacts with the IgA heavy chain and the other (helix-b) using equivalent residues to contact solvent and a smaller interface with IgA. Prior studies, including mutational analysis, identified M4 residues 45–73 as the minimal IgA-binding region[9]; our

observations indicate that residues 45, 70, 71, and 73 contribute indirectly by stabilizing the M4 coiled-coil dimer.

### The structure of the SIgA-M4 interface
Despite the reported accessibility of two FcαR binding sites on SIgA, our structure reveals M4 bound only to $Fc_{AB}$[13]. The M4 interface includes 11 residues on the Cα3 domain of both heavy chain-A (HC-A) and heavy chain-B (HC-B) (Fig. 3), with a total buried surface area of

1000 Å$^2$. Participating IgA residues are predominantly located along one β-strand in HC-A (F443, T444, Q445, T447), forming hydrogen bonds with M4 helix-a residues R49 and E54. HC-A E389 and R346 also reach out to interact with M4 helix-a residues Q47 and D57, respectively, and HC-A D449 forms a salt bridge with R59 on helix-b of M4. Additionally, HC-A Tp residue K454 may mediate longer-range electrostatic interactions (e.g., 4.5 Å) with M4 helix-b residues D67 and E70. The M4 interface with IgA HC-B includes only residues S356, E357 and A360, which form hydrogen bonds with Q66 of helix-b and R61, K68 of helix-a, respectively. All IgA interface residues are conserved in human IgA subtypes (IgA1, IgA2m1, IgA2m2) and most are conserved in mouse IgA, with the exception of R346 and L441 (Supplementary Fig. 2). Mouse IgA is not known to bind M4 thus R346 and L441 appear to be especially important for M4 binding and structurally equivalent residues in mouse IgA, P341 and M436 respectively, do not appear to support the interaction[33]. Comparing M4 and M22 sequences, we find seven out of 11 IgA contact residues are identical and the other four residues are similar (Fig. 2b), consistent with the prior analysis that identified similar IgA-binding motifs in M4 and M22[9].

## Mutagenesis confirms the key binding residues on M4

Although side chain density was apparent in the structure, the N and C termini of the M4 coiled-coil were not resolved, and therefore to validate residue assignments and investigate the contributions of individual M4 residues on SIgA binding, we conducted structure-based mutational analysis and surface plasmon resonance (SPR) binding assays. Based on the pairwise interactions between M4 and SIgA (Fig. 3b) and pairwise interactions between M4 helix-a and helix-b, we made three individual M4 point mutations Q47A, R49A, and K68A. A subset of residues, including helix-b Q66, are positioned such that side chain atoms could interact with both IgA and the other helix (e.g., helix-a); these were not mutated due to the possibility of disrupting the M4 dimer. M4 and mutant variant binding to SIgA were determined by SPR. Sensorgrams revealed lower responses for all M4 mutants when compared to concentration matched wild-type M4, as well as lower dissociation constants (Fig. 3c and Supplementary Fig. 3). Compared to wild-type M4 ($K_D = 20$ nM), mutant variants R49A ($K_D = 159$ nM) and K68A ($K_D = 156$ nM) exhibited the most pronounced reduction in IgA-binding, consistent with the observation that R49 interacts with one of the human-IgA specific binding site residues, L441. K68 interacts with HC-B, indicating that contacts with both HC-A and HC-B chains are important.

## The structure of SIgA-CD89 complex

The unexpected stoichiometry of the SIgA-M4 complex raised the question of how host FcαRs bind SIgA. Modeling based on the structure of mIgA in complex with CD89 and SIgA structures suggested that two CD89 can bind SIgA[14]; however, to verify this possibility, we determined the Cryo-EM structure of SIgA with the CD89 ectodomain to a resolution of 3.2 Å (Fig. 4a). The ectodomain of CD89 consists of two immunoglobulin-like domains (D1 and D2) arranged at an angle of approximately 90° to each other. Map quality and resolution were sufficient to refine the positions of main chain and side chain atoms for the residues at the interface, while density in the distal region of CD89, including the second Ig-like domain D2, was of lower resolution. The refined structure revealed a 2:1 CD89:SIgA binding stoichiometry verifying that two copies of CD89 can bind both mIgA and SIgA (Fig. 4a). The two copies of CD89 bound to SIgA are separated by 108 Å (distance between the two copies of D2 C-terminal residue T195) and relative to each other are oriented differently than the two CD89 bound to mIgA for which C-terminal residues are reportedly spaced by 124 Å (Fig. 4a, c)[2]. Alignment to the mIgA-CD89 structure revealed superimposable binding sites (RMSD = 1.165 at Fc$_{AB}$; RMSD = 1.121 at Fc$_{CD}$) on Cα2 and Cα3[2]. Consistent with the mIgA-CD89 structure, the

CD89 D1 BC loop, the D strand, the DE loop, and the FG loop (Ig-like domain nomenclature) contacted a total of 19 IgA residues, including three Cα2 residues and 16 Cα3 residues, burying a total of 1656 Å$^2$ (Fig. 4b).

## CD89 and M4 share overlapping but distinct SIgA-binding sites

A comparison of the CD89 and M4 binding footprints on SIgA revealed five shared interface residues (E389, L441, F443, T444, Q445) located on the Cα3 domain. These residues are among those included in the Cα2/Cα3 "hot spot" for receptor and bacterial protein binding. CD89 binds additional residues located toward and on Cα2, whereas M4 contacts only Cα3 and binds residues closer to the Tps and JC, including R346, T447, D449 from HC-A and S356, E357, A360 from HC-B (Fig. 5a). CD89 and M4's distinct binding footprints and structures direct CD89 D2 C-terminal residues away from SIgA and direct the M4 fragment C-terminal residues (and likely the rest of the M4 coiled-coil), alongside Fc$_{AB}$ in a line near parallel to Cα3 (Fig. 5b). Noted differences contribute to the distinct stoichiometry of CD89 versus M4 binding to SIgA. In the case of M4, the inclusion of S356, E357, A360 and R346 in the M4 binding interface, together with its extended coiled-coiled topology, enforce the surprising 1M4:1SIgA stoichiometry because the steric accessibility of these four residues and neighboring residues differs between Fc$_{AB}$ and Fc$_{CD}$. In Fc$_{AB}$, M4 can bind S356, E357, A360 and R346 with residues on both helices and extend the coiled-coiled over neighboring residues and toward the Tps. In Fc$_{CD}$, S356, E357, A360 and R346 remain solvent accessible, but JC loop residues 24–32 bind the adjacent region, blocking the path occupied by M4 (helix-b residues 69–72) and therefore occluding binding (Fig. 5b). The JC loop that occludes M4 binding does not interfere with CD89 binding on Fc$_{CD}$, implying that sterically, CD89 and M4 ectodomains could bind to the same copy of SIgA, with M4 binding Fc$_{AB}$ and CD89 binding Fc$_{CD}$ (Fig. 5c); however, whether this would occur in the context of bacterial and host-cell interactions remains to be determined.

## Discussion

IgA is associated with diverse effector functions mediated by monomeric and secretory forms. Both forms have been shown to bind human CD89 and GAS; yet, interactions with SIgA and potential differences between mIgA and SIgA engagement are poorly understood, limiting knowledge of IgA effector functions and host-pathogen interplay[16,34]. Host FcαRs and bacterial IgA-binding proteins have long been appreciated to bind mIgA through the conserved "hot spot", however only recently were hot spot residues shown to be accessible on SIgA[6,11,12]. Our work demonstrates that CD89 and M4 (and likely M22) can engage SIgA through just five common residues and exhibit distinct binding footprints and stoichiometry, implying that while the hot spot is conserved on mIgA and SIgA, its accessibility to different receptors and proteins is variable and has potential to influence functional outcomes and provide unique selective advantages for both the host and the pathogen.

It is notable that both mIgA and SIgA can bind two copies of CD89 in vitro but that the spacing and orientation of bound CD89 are different in CD89-mIgA and CD89-SIgA complexes. Earlier reports suggested that the 124 Å distance separating the CD89 D2 domains is too large to initiate signaling (when two CD89 bind the same copy of mIgA)[2]. This may also be the case for SIgA, in which we find the two copies of CD89 to be separated by 108 Å (previously modeled to 99 Å apart), although additional studies are needed to evaluate CD89 function in a host-cell membrane[15]. If this is the case, then CD89 signaling will occur when copies of CD89 bind adjacent IgA on IgA-antigen complexes. We envision that each monomeric/polymeric form of IgA (and its secretory forms) may provide a unique set of CD89-binding sites, each with different accessibility and each oriented differently relative to bound antigen and relative to copies of CD89 on the cell membrane. This, in turn, may influence CD89 clustering, potential for signaling, and

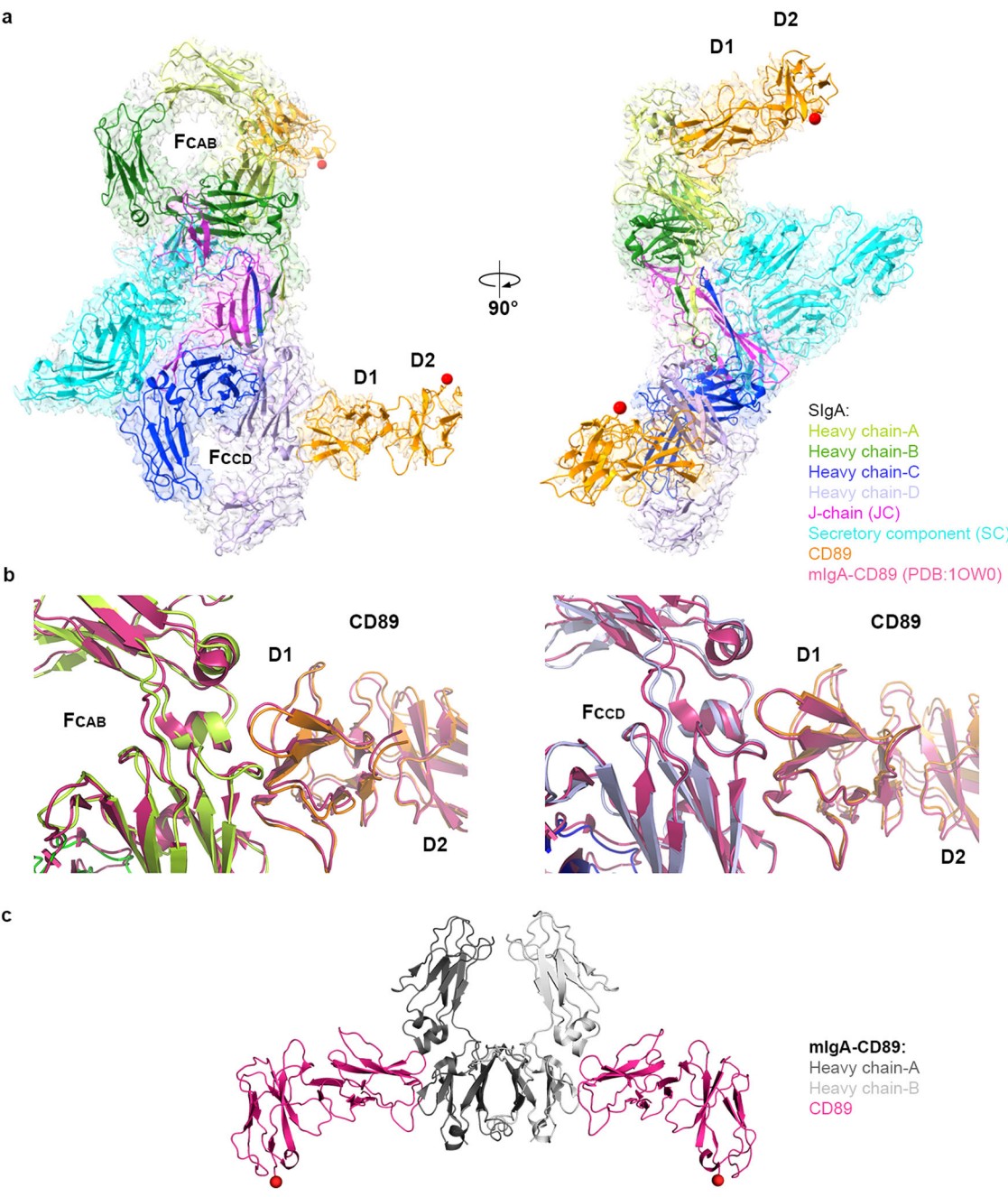

**Fig. 4 | SIgA-CD89 structure. a** The Cryo-EM structure of CD89 in complex with human SIgA is shown as a cartoon with a partially transparent map in two orientations. The antigen-binding fragments (Fabs) are disordered. The two Ig-like domains of CD89 are labeled D1 and D2. CD89 D1 contacted a total of 19 IgA residues, including Cα2 residues L256, L257, L258 and Cα3 residues E348, R382, L384, S387, E389, M433, H436, E437, A438, L439, P440, L441, A442, F443, T444, Q445. **b** Alignment of the mIgA-CD89 crystal structure (PDB: 1OW0) to the Fc_AB-CD89 interface (left) and Fc_CD-CD89 interface (right) on SIgA-CD89. The BC, DE and FG loops are labeled on Fc_AB-CD89. **c** Crystal structure of CD89 in complex with human Fcα (PDB code: 1OW0). All models are colored according to the keys. The C-termini of CD89 D2 domains are indicated by red spheres, which on CD89-SIgA are 108 Å apart.

effector functions. Speculatively, this provides an explanation for how different forms of IgA in different locations (e.g., serum or mucosa) may elicit different outcomes upon encountering CD89, or perhaps other FcαRs such as human Fcα/µR, and why additional factors (e.g., Mac-1) might be involved in SIgA-CD89-dependent signaling[5,35].

Whereas mIgA and SIgA share the same number of CD89-binding sites, this does not appear to be the case for M4 (and presumably, M22) since the JC is not present in mIgA. Because differences in mIgA and SIgA engagement may signify different functional outcomes from host-GAS interactions, we modeled the M4 structure onto Fcα and tested soluble M4 binding to mIgA using SPR. Modeling verified the steric accessibility of two M4 binding sites, and SPR experiments revealed similar dissociation constants, ~15 nM and 20 nM, for soluble M4 binding to mIgA and SIgA, respectively (Supplementary Fig. 5). This suggests that the two M4 binding sites on mIgA are molecularly equivalent to the single site on SIgA. These observations are consistent with publications reporting similar affinities for M22 binding to mIgA and SIgA and those reporting similar levels of mIgA and SIgA binding to GAS cultures[16,36]. However, the binding of two copies of M4 to one mIgA could enhance the avidity of IgA-GAS interactions, effectively strengthening the interaction while also completely blocking CD89-binding sites. This model is consistent with other reports indicating

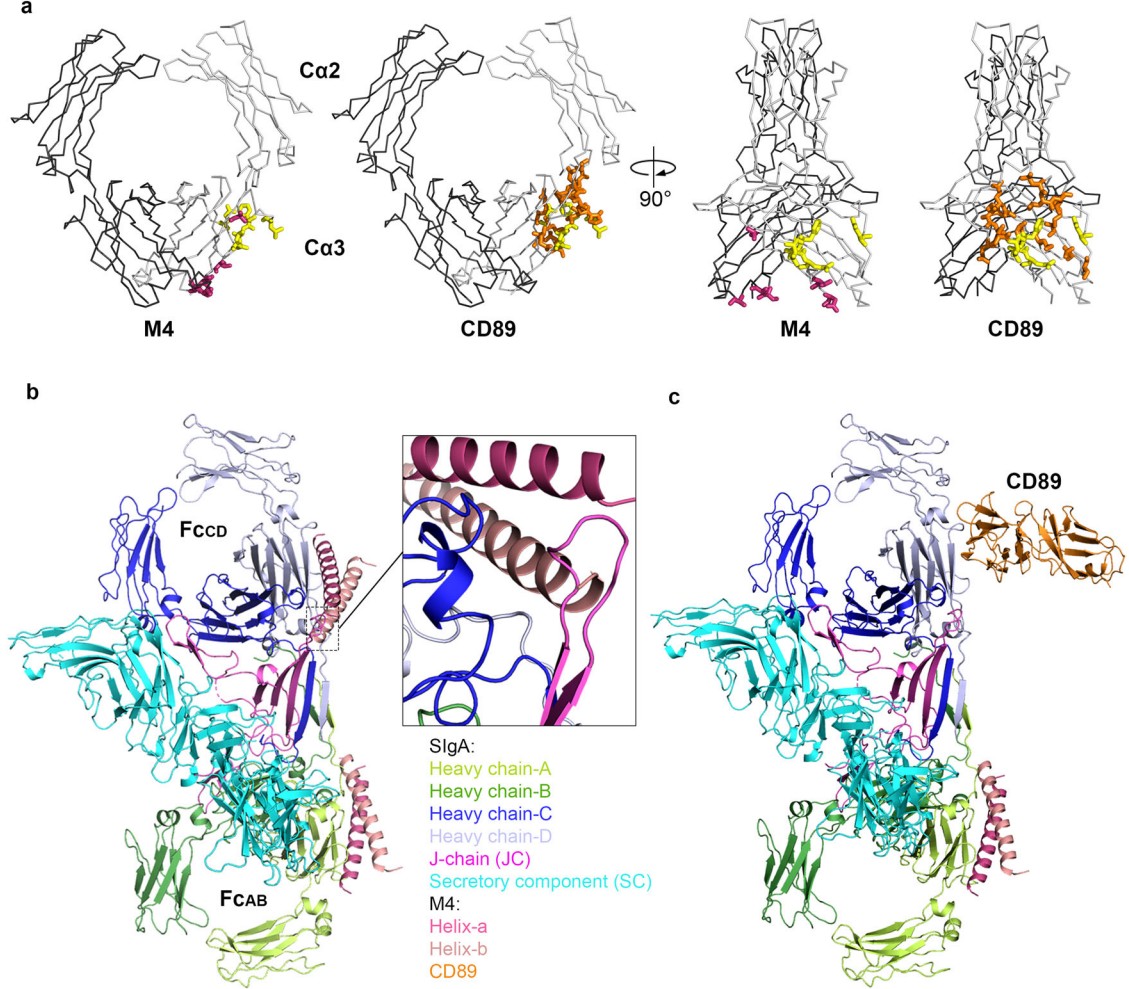

**Fig. 5 | M4 and CD89-binding site comparison and modeling. a** Comparison of M4 and CD89-binding sites on Fcα shown in two orientations. Fcα is shown as ribbons, and binding residues are shown as stick representation. The 11 residues bound by M4 are shown in pink and yellow and the 19 residues bound by CD89 are shown in orange and yellow. In all panels, five shared binding site residues, E389, L441, F443, T444, Q445, are colored yellow. **b** M4 modeled at the second site (Fc$_{CD}$; blue) is partially overlapping with JC (magenta), indicating a steric clash. **c** Model of M4 (pink) and CD89 (orange) bound SIgA. CD89 bound to the Fc$_{CD}$ site does not clash with SIgA residues.

that M4 binds mIgA with higher affinity than SIgA and with proposals that M4 functions to block CD89 effector functions[11]. However, whether each mIgA bound to GAS can, in fact, be bound by two copies of M4 in vivo remains to be tested; it is also unclear how many CD89-binding sites need to be blocked to inhibit effector functions.

In the case of SIgA interactions with GAS, the sterically enforced 1:1 M4:SIgA stoichiometry is intriguing because SIgA adopts an asymmetric structure, which published modeling predicts will influence the positions of Fabs and accessibility of the SC and FcαR binding sites[13]; therefore, M4 binding to Fc$_{AB}$ positions SIgA differently than if it could bind to Fc$_{CD}$. To visualize this possibility, we used Alphafold2-multimer to model the full-length M4, aligned it to our M4-SIgA structure and created schematic representations of the complex on a bacterium surface (Fig. 6)[37]. Assuming that M4 is oriented approximately perpendicular to the outer surface of the membrane, it would extend ~400 Å, placing the bound SIgA ~250 Å from the surface where one set of Fabs would be directed upward and away from the bacterium surface and the other set directed toward the surface. In this model, the SC protrudes upward and away from the bacterium surface, similar to the M4 HVR, and the Fc$_{CD}$ binding site is located closer to the bacterium surface and approximately 62 Å from M4 (Fig. 6b). Other GAS surface proteins are expected to be shorter than M4, implying that bound SIgA would occupy a position near the outermost surface of the

bacterium[38]. While information describing M4 flexibility, as well as the density of M4 and other surface antigens, is needed to fully evaluate the geometric arrangement of SIgA bound to the bacterial surface, our data and analysis indicate that the 1:1 M4:SIgA stoichiometry will constrain the location and orientation of bound SIgA, a possibility that should be considered when exploring functional outcomes of bacterial-SIgA interactions (Fig. 6).

How might SIgA binding to GAS M4 (or M22) in a constrained orientation influence the host or microbe? One possibility is that M4 binding to Fc$_{AB}$ impacts the binding of SIgA Fabs to other antigens on the surface of GAS, such as fibronectin-binding proteins (SFb), *Streptococcus pyogenes* cell envelope protease (SpyCEP), streptococcal C5a peptidase (Scp)[39]. Bacteria that can bind SIgA through M4 (or M22), may be able to limit antibody binding to other surface proteins, although, as suggested by others, the relative affinity of antibodies to other surface antigens and their steric accessibility will likely impact the outcome[40]. Another possibility is that SIgA-bound M4 influences the localization of the bacterium (e.g., in host mucosal secretions). SIgA is covered in host glycans, which have been implicated in binding to host mucus, lectins, and even microbial proteins, something we speculate could be promoted by the position of the highly glycosylated SC[41,42]. Our model predicts the D2 domain of SC, which is the most accessible among the five domains, would be exposed near the M4 HVR toward the outermost

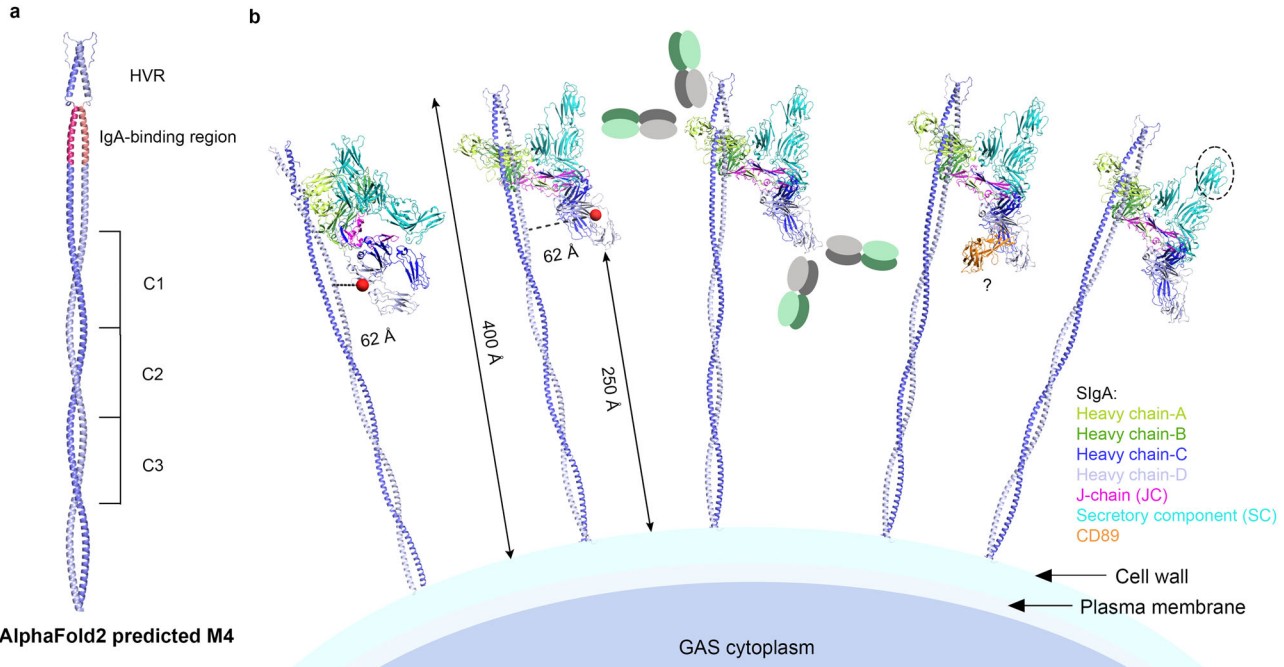

**Fig. 6 | Schematic model of human SIgA binding to M4 on a GAS surface.**
**a** AlphaFold2-multimer predicted M4 dimer; residues 287–316 are predicted to be disordered and not shown. The IgA-binding region on the predicted model is colored in pink and has an RMSD of 0.640 when aligned to the SIgA-M4 structure. **b** Surface-associated M4, shown in two orientations, binds to one side of SIgA (Fc_AB), holding all SIgA at a similar orientation and distance from the bacterium surface while leaving the Fc_CD FcαR binding site (red sphere) open. Possible locations of disordered Fabs are shown as gray and green ovals. CD89 binding to M4-bound SIgA is sterically possible and modeled on one copy of the SIgA-M4 complex. SIgA SC domain D2 is indicated by a circle and is exposed near the outermost surface of the bacterium.

surface of the bacterium where similar to other SIgA (not bound to GAS), it might mediate interactions with other factors (Fig. 6)[13].

The 1:1 M4:SIgA stoichiometry also leaves open the possibility that M4-bound SIgA could bind CD89 (or another FcαR such as Fcα/μR) at the Fc_CD site and influence effector functions in some way (Fig. 6b). The accessibility of these sites on mIgA and SIgA has not been verified on a bacterial surface and it seems plausible that CD89 activity could be inhibited by occluding just one of the two CD89-binding sites, especially since our model predicts that the Fc_CD site on SIgA-M4 complexes would be pointed downward toward the bacterial membrane, potentially occluding the site. In this case, SIgA may act like mIgA, restricting CD89-dependent effector functions. Still, we cannot rule out the possibility that a sterically accessible Fc_CD binding site and kinetics that favor CD89 binding might allow one copy of CD89 to bind Fc_CD and, subsequently, another copy to compete with M4 for the Fc_AB site. Another possibility is that CD89 binding to Fc_CD could promote other copies of CD89 binding to local SIgA (and/or mIgA) that is not bound by M4 but rather is bound to another antigen with its Fabs.

The idea that IgA and its effectors have provided selective pressure driving GAS evolution is reasonable given the existence of at least seven M protein virulence factors that bind IgA and the prevalence of M4 and M22 strains, which are abundant and geographically widespread[22,23,29,43]. Interestingly, both M4 and M22 strains lack a Hyaluronic Acid (HA) capsule, found on other GAS strains, which is antiphagocytic and has long been proposed as an essential virulence factor[44,45]. We speculate that the Ig-binding proteins provide alternative and compensatory mechanisms for M4 and M22 strain virulence. Regardless, our work raises the possibility that SIgA has shaped GAS evolution, perhaps providing different selective pressures than mIgA, and also highlights the importance of further investigating GAS-SIgA interactions in the mucosa where M4/M22 might block host FcαR effector functions and/or block antibody binding to surface antigens and/or provide another survival advantage for the bacteria (e.g., adhesion to mucosal factors).

The notion that host SIgA effector functions play a significant role in antimicrobial response and may provide selective pressure extends beyond GAS, given that at least three distinct pathogenic bacterial species shown to bind SIgA express specific IgA and/or SIgA-binding proteins, including *S. pyogenes*, *S. aureus* and *S. pneumoniae*, all of which utilize the nasopharynx as their primary human reservoir where encounters with SIgA have potential to influence virulence and host response[46,47]. To explore potential similarities and differences between M4 and other IgA-binding proteins, we modeled a *S. aureus* SSL7-SIgA complex by aligning the crystal structure of SSL7-Fcα (PDB:2QEJ) to each Fc on SIgA (Supplementary Fig. 6)[7]. Resulting models suggest that SSL7 can bind both the Fc_AB and Fc_CD "hot spots," resulting in 2:1 SSL7:SIgA stoichiometry, as observed for CD89:SIgA. This stoichiometry appears possible because SSL7 adopts a flat "disk-like" topography that, when bound to IgA or SIgA, positions the C-terminus away from Fcα, similar to CD89 and in contrast to M4[7]. This structure and trajectory likely allow SSL7 to bind SIgA Fc_CD without being occluded by the JC loop as observed for M4, signifying that among Fcα-binding proteins, the sterically enforced 1:1 M4:SIgA stoichiometry appears to be unique to M4 and hinting at functional differences relevant to understanding host interactions with GAS and *S. aureus*. Unlike SSL7 and M4, which bind the IgA Fc, the *S. pneumoniae* SpsA (also called CbpA) protein binds directly to pIgR or SC, meaning that it interacts only with SIgA and not mIgA. This further supports the notion that microbial-SIgA interactions are functionally and broadly relevant; together structural data provided here and in other recent reports provide a framework to further elucidate the consequences of those interactions for both the host and the microbe[2,6,13–15,48].

## Methods

### Protein expression and purification
Genes encoding the human IgA1 heavy chain constant region and the lambda light chain constant region were fused with STA121 VH and VL

domain sequences[49]. The human tissue plasminogen activator (tPA) signal sequence was inserted upstream of the IgA heavy chain sequence. These sequences, along with human JC and SC, were codon optimized, synthesized (IDT) and each cloned into mammalian expression vector pD2610v1 (Atum). All constructs were transiently co-transfected into HEK Expi293F (Gibco: A14527) cells using ExpiFectamine to produce the SIgA complex (Thermo Fisher). Six days after transfection, SIgA was purified from the cell supernatant using CaptureSelect™ Human IgA Affinity Resins. Superose 6 (Cytiva) size exclusion chromatography (SEC) was used to further purify SIgA. The gene encoding the human CD89 ectodomain (uniprot ID: P24071) was fused to a C-terminal hexahistine-tag, codon-optimized, synthesized (IDT) and cloned into pD2610v1. The CD89 construct was transiently transfected into HEK Expi293F cells using ExpiFectamine. CD89 was purified from the cell supernatant using Ni-NTA affinity resin (Qiagen). Superose 6 SEC was used to further purify CD89. The gene encoding the *Streptococcus pyogenes* M4 protein (uniprot ID: P13050) was codon optimized, synthesized (IDT) and cloned into pET28_a (+) vector (Novagen). The M4 endogenous N-terminal signal peptide and C-terminal cell wall sorting signal were not included in the construct and a C-terminal hexahistine-tag was added. M4 mutants Q47A, R49A, K68A were generated using the Q5 site-directed mutagenesis kit (NEB, E0554). The M4 mutant variant constructs were transformed and expressed in *E. coli* strain BL21 (DE3); expression was induced using 1 mM IPTG, added to the culture growing at 37 °C when OD reaches 0.4–0.6. Cells were harvested 3–5 h after IPTG induction by centrifugation, resuspended in 50 mM Tris-HCl, 150 mM NaCl, pH 7.5, 1 mM phenylmethylsulfonyl fluoride and lysed by sonication. The M4 was isolated from a soluble fraction of cell lysate using Ni-NTA affinity resin. Superose 6 SEC was used to further purify M4. Purified SIgA and M4 or CD89 were mixed in a 1:2 molar ratio and incubated in storage buffer (50 mM Tris-HCl, 150 mM NaCl, pH 7.5) at 4 °C overnight, and complexes were purified using Superose 6 SEC.

### Cryo-EM grid preparation and data collection

SIgA-M4: Quantifoil R2/2 300 mesh holey carbon grids were glow discharged for 60 s; 3 µl of SIgA-M4 at 1.0 mg/ml was applied to each grid with Vitrobot Mark IV at SLAC with 1 blot of 1.5 s, 0 offsets, 5 s wait, at 100% humidity and 4 C°. Movies were collected using EPU 2.1.0 on a Titan Krios (TEMALPHA at SLAC) operating at 300 kV and with a Gatan K4 direct electron detector. Both untilted and 25° tilted movies were collected at 0.95 Å/Pixel, 40 frames per movie, total exposure time of 4.95 s and total dose of 50 electrons/Å$^2$.

SIgA-CD89: Quantifoil R1.2/1.3 400 mesh holey carbon grids were glow discharged for 60 s; 3 µl of SIgA-CD89 at 1.0 mg/ml was applied to each grid with Vitrobot Mark IV with 1 blot of 2 s, 2 offset, 5 s wait, at 100% humidity and 4 C°. Movies were collected using SerialEM 4.0 on a Titan Krios at Purdue University operating at 300 kV and with a K3 direct electron detector. Both untilted and 30° tilted movies were collected at 0.82 Å/Pixel, 60 frames per movie, total exposure time of 2.41 s and total dose of 71.3 electrons/Å$^2$. Data collection statistics can be found in Supplementary Table 1.

### Cryo-EM data processing

SIgA-M4: 2600 movies were collected with untilted stage and 3251 movies were collected with stage tilted at 25°. All data processing were done in CryoSPARC v.3.2.0[50]. Initial processing of the two data sets was done separately. Movies were motion corrected, CTF estimated and manually curated based on ice thickness, total motion and CTF fits, resulting in 2081 untilted and 2786 tilted movies. Particles were picked using a blob picker in CryoSPARC with minimum and maximum particle diameter of 100 Å and 200 Å. Particles were extracted with 2x binning. 2D classification of these particles generated initial references for template-based particle picking, which gave 970k particles from untilted movies and 1.3 million particles from tilted movies. Those particles were combined and subjected to several rounds of 2D

classification to remove bad classes. A final set of 200 K particles was used to generate the ab initio model. Particles in the final set were then re-extracted at full-pixel size and used for non-uniform refinements of the model. The final refinement generated a map with an average resolution of 3.07 Å at FSC = 0.143.

SIgA-CD89: 2955 movies were collected with untilted stage and 3696 movies were collected with stage tilted at 30°. All data processing were done in CryoSPARC v.4.1.2[50]. Initial processing of the two data sets were done separately. Movies were motion corrected, CTF estimated and manually curated based on ice thickness, total motion and CTF fits, resulting in 2865 untilted and 3105 tilted movies. Particles were picked using blob picker in CryoSPARC with minimum and maximum particle diameter of 150 Å and 250 Å. Particles were extracted with 2x binning. 2D classification of these particles generated initial references for template-based particle picking, which gave 507 K particles from untilted movies and 732 K particles from tilted movies. Those particles were combined and went through several rounds of 2D classification to remove bad classes. A final set of 249 K particles was used to generate the ab initio model. Particles in the final set were then re-extracted at full-pixel size and used for non-uniform refinements of the model. The final refinement generated a map with an average resolution of 3.18 Å at FSC = 0.143.

### Structure building and refinement

SIgA-M4: The structure of human SIgA (PDB: 6UE7) was docked into the map using UCSF Chimera 1.15[51]. The M4 fragment was hand-built using the Coot Molecular Graphics Package 0.8.92[52]. All chains of SIgA and M4 were refined as rigid bodies using Phenix (1.20.1) real space refinement[53]. Subsequent iterations involved inspection of the map-model fit and manual adjustment of the model followed by Phenix real space XYZ refinement. Iterations of Phenix refinements and manual adjustments were repeated until model quality no longer improved.

SIgA-CD89: Human SIgA structure (PDB: 6UE7) and mIgA:CD89 structure (PDB:1OW0) were docked into the map using UCSF Chimera[51]. All chains of SIgA and CD89 were refined as rigid bodies using Phenix real space refienemnt[53]. Inspection of the map-model fit and manual adjustment of the model were done after each Phenix real space XYZ refinement. Iterations of Phenix refinements and manual adjustments were repeated until model quality no longer improved. Model and refinement statistics can be found in Supplementary Table 1.

### Surface plasmon resonance

Surface plasmon resonance (SPR) binding experiments were done using a Bruker Sierra SPR-32 Pro instrument at room temperature. Next, 0.2 uM human mIgA or SIgA in sodium acetate buffer (pH = 4.5) was coupled to lanes B and D, respectively, on a high-capacity amine (HCA) chip using an amine coupling kit (Bruker 1862634). Lanes A and C were mock coupled (with only sodium acetate buffer pH 4.5) and used as references. Responses were measured for a two-fold dilution series of each analyte (M4 wild type and mutants Q47A, R49A, K68A) in HBS-EP+ buffer (0.01 M HEPES pH 7.4, 0.15 M NaCl, 3 mM EDTA, 0.005% v/v Surfactant P20) with a high concentration of 0.25 uM. All binding experiments were done with 120 s contact time, 180 s dissociation time, and flow rate of 25 µl/min. Surfaces were regenerated with 2.5 M MgCl$_2$. All data were collected using the Sierra SPR Control software. Data were fitted to 1:1 binding models, and associated binding constants were calculated using Bruker Analyzer R4 software.

### Structure modeling, analysis, and sequence alignment

Human SIgA (PDB: 6UE7) was aligned to the M4-bound SIgA in The PyMOL Molecular Graphics System (Version 2.5.1 Schrödinger, LLC), and the RMSD between the bound and unbound state of SIgA was calculated automatically during the alignment performed by PyMOL (A>align>align to molecule). A list of all possible interactions between SIgA and M4 was generated using the protein interaction calculator

(PIC) webserver[54]. The interactions were visualized and manually checked in PyMOL and contacts within 4 Å were selected and reported in Fig. 3b. The buried surface area was calculated using the 'get_area' function in PyMOL.

AlphaFold2-multimer was used to predict the full-length M4 dimer structure and ColabFold 1.5.2 was used to generate the prediction[37]. M4 residues 286–313 are predicted as disordered and not shown in figures. Full-length modeled M4 residues 45–73 were aligned to the corresponding residues in the SIgA-M4 structure using PyMOL, with an RMSD of 0.640. The aligned structure of full-length M4 with SIgA were used in Fig. 6. The distance of M4 to GAS surface was estimated by measuring the distance between M4 residues 15 and 285. The distance from the FcαR site on Fc$_{CD}$ to M4 was measure from HC-D residue L441 to M4 helix-b residue H97.

The sequence alignment between human and mouse IgA heavy chains (human: Uniprot P01876, P01877, A0A0G2JMB2; mouse: Uniprot P01878) was carried out using ClustalOmega and the alignment figure was made using ESPript3[55,56]. The alignment of mIgA:CD89 (PDB: 1OW0) to SIgA:CD89 and RMSD calculation were done in the same way as described above.

### Figures
Structural figures were made using UCSF Chimera1.15[51] and the PyMOL Molecular Graphics System (Version 2.5.1 Schrödinger, LLC); SPR data were plotted using Bruker Sierra-32 Analyzer and Microsoft Excel[51]. All figures were assembled using Adobe Photoshop.

### Reporting summary
Further information on research design is available in the Nature Portfolio Reporting Summary linked to this article.

## Data availability
The Cryo-EM density maps have been deposited in the EM databank (www.ebi.ac.uk/emdb) with the accession codes EMD-40568, and EMD-40567 for M4-SIgA and CD89-SIgA structures, respectively, and the refined coordinates have been deposited in the Protein Data Bank (www.rcsb.org) with accession codes 8SKV (M4-SIgA) and 8SKU (CD89-SIgA). The human SIgA structure (PDB code: 6UE7) and CD89-Fcα structure (PDB code: 1OW0) were used as initial models for structure building and modeling. The CD89-Fcα structure (PDB code: 1OW0) was used for comparative structure analysis and the SSL7- Fcα structure (PDB code: 2QEJ) was used for modeling in Supplementary Fig. 6. The SPR data generated in this study are provided in the Source Data file. Source data are provided with this paper.

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

## Acknowledgements

This work was supported by NIH R01 AI165570 (PI Beth Stadtmueller), University of Illinois start-up funding and the Michael A. Recny Graduate Fellowship (awarded to Q.L.). Cryo-Electron microscopy data were col-lected at the Purdue Cryo-EM facility (http://cryoem.bio.purdue.edu) with assistance from Thomas Klose and the Stanford-SLAC Cryo-EM Center (S$^2$C$^2$) supported by the National Institutes of Health Common Fund Transformative High Resolution Cryo-Electron Microscopy pro-gram (U24 GM129541) with assistance from Htet A. Khant. We thank members of the Stadtmueller lab for insightful discussion and Sarah Leonard for supporting SPR experiments.

## Author contributions

The study was conceived by B.M.S. and Q.L.; experiments were con-ducted by Q.L.; B.M.S. and Q.L. analyzed data and wrote the manuscript.

## Competing interests

The authors declare no competing interests.

## Additional information

**Peer review information** *Nature Communications* thanks Andrew Herr, Gunnar Lindahl and the other anonymous reviewers for their contribu-tion to the peer review of this work. A peer review file is available.

