## [Peer Review File · Nature Communications]

The Structures of Secretory Immunoglobulin A in complex with Streptococcus pyogenes M4 and human CD89REVIEWER COMMENTS

Reviewer #1 (Remarks to the Author):

Secretory IgA (SIgA) protects against infections at mucosal surfaces and has a more complex structure than monomeric IgA (mIgA), which is found in serum and has poorly understood function. For both forms of IgA, the human phagocyte receptor CD89 binds to the Fc part, an interaction that may promote phagocytosis *in vivo*. Moreover, both forms of IgA bind streptococcal IgA-Fc-binding surface proteins, which target a site that overlaps with the CD89-binding site. Special interest has been devoted to IgA-binding streptococcal M proteins, which belong to a family of coiled-coil proteins essential for virulence in the human pathogen *S. pyogenes*, a common cause of throat and skin infections. Because IgA-binding M proteins such as M4 inhibit the binding between mIgA and CD89, it has been suggested that these M proteins interfere with IgA-mediated phagocytosis. However, little is known about the *in vivo* function of IgA-binding M proteins.

While earlier studies were focused on mIgA, Liu and Stadtmueller studied the interaction of CD89 and M4 with SIgA, arguing that such studies are particularly relevant, because SIgA has a key role in preventing infections at mucosal surfaces. Cryo-EM of complexes convincingly demonstrated that the stoichiometry was 1:2 for the SIgA-CD89 complex, as expected from modeling, but for the SIgA-M4 complex it was unexpectedly found to be 1:1, implying that only one of the predicted M4 binding sites is accessible in SIgA. A likely explanation is that the J chain in SIgA sterically interferes with binding of M4, but not CD89, to the second region. This interesting result sheds new light on the binding properties of SIgA and may have implications for the poorly understood role of streptococcal IgA-binding proteins. Specifically, the data suggest that M4 may not interfere with phagocytosis promoted by interaction between SIgA and CD89. However, it is very uncertain whether SIgA plays a significant role in the defense against local *S. pyogenes* infections (see below), so the biological relevance of the findings is not clear. Overall, the intriguing molecular data reported in this ms would benefit from being considered in the context of *S. pyogenes* infections. In any case, it is of much interest that this ms reports the first structure of an M protein-IgA complex, with verification of structural data by site-specific mutagenesis.

Major comments:

1) The authors conclude that SIgA binds two molecules of CD89 but only one molecule of M4, implying that a complex M4-SIgA-CD89 can be formed even in the presence of an excess of M4. This conclusion is based exclusively on comparisons of the two cryo-EM structures and should be substantiated by other means, e.g., by studying the binding properties of SIgA bound to immobilized M4 or by the characterization of soluble complexes.

2) The authors stress the difference between the situation in blood, where *S. pyogenes* would encounter mIgA, and the situation on mucosal surfaces, where the bacteria would encounter SIgA. Since *S. pyogenes* only rarely enters blood, it may appear reasonable to focus on the interaction with SIgA. However, a local infection, such as a 'strep throat', will trigger an inflammatory response that

results in massive exudation of plasma proteins and recruitment of neutrophils. Thus, the bacteria will encounter mIgA and phagocytes in the inflamed tissue, potentially allowing proteins such as M4 to interfere with mIgA-mediated phagocytosis. In contrast, it is uncertain whether SIgA plays a role in this situation. This argument becomes even stronger when it comes to skin infections (impetigo), since *S. pyogenes* will only encounter exudated mIgA in that case. Of note, impetigo is the most common *S. pyogenes* disease in many parts of the world and strains that bind IgA-Fc typically cause infections of the skin, not the throat, so strains that bind IgA may in most infections not encounter SIgA. It is also noteworthy that M4 binds to mIgA with an affinity that is more than 10-fold higher than for SIgA (ref. 26), suggesting that M proteins have evolved to selectively bind mIgA. The situation is even more striking for the *Streptococcus agalactiae* (GBS) protein beta (Bac), which has high affinity for mIgA but binds SIgA only weakly. While these arguments focus interest on the interaction between pathogenic streptococci and mIgA, they do not exclude that the features of SIgA described by Liu and Stadtmueller play a role in certain situations, although the available evidence indicates that mIgA is biologically more relevant in *S. pyogenes* infections.

Other comments:

- 3) Discussion: It is very long, so the authors may consider removing the SSL7 text (lines 242-257), which is interesting but does not fit well into the paper. -- Because the interaction of M4 with SIgA may be of limited importance, as compared with the binding to mIgA (see above), it becomes problematic that the Discussion puts such strong emphasis on this interaction. -- The argument on lines 315-319 is unclear. -- Concerning the role of IgA (mIgA or SIgA) bound to an M protein, the authors may consider the possibility that it is unrelated to the interaction between IgA and CD89.
- 4) The numbering of aa residues in the IgA-binding regions of M4 and M22 is permuted one step, as compared to the correct one (see refs. 6 and 5). This must be corrected throughout, in text and figures.
- 5) line 35: "...interactions that allow...". A more cautious phrasing would be "...interactions that may allow...".
- 6) lines 66-68: It is correct that *S. pyogenes* tonsillitis is much more common than invasive infections, but it is invasive infections, not tonsillitis, that cause over 160,000 deaths per year.
- 7) line 84: the region at the N-terminus of an M protein is not the most immunogenic region but the key site for protective antibodies (Lannergård et al., Cell Host Microbe 2011).
- 8) lines 88-89: the words 'including five C-terminal residues of the HVR' are unclear and can be removed.
- 9) line 105: '...in contrast to previously proposed...'. Already in ref. 24, it was reported that the (overlapping) sites interacting with the CD89 and M4 are nonidentical.
- 10) line 236: 'prior mutational analysis'. To my knowledge, such analysis has not been reported. Thus, the word 'mutational' can be deleted.
- 11) line 267: the words 'outer membrane' should be removed. Since *S. pyogenes* is a Gram-positive bacterium, it does not have an outer membrane.
- 12) line 281: "...surface antigens, such as..." Most of the examples given are not surface proteins, but proteins released from the bacteria.

13) line 351: Like other M proteins, M4 has a signal peptide but does not have a ‘propeptide’

Reviewer #2 (Remarks to the Author):

This manuscript describes separate cryoEM structures of SigA bound to a streptococcal immune evasion protein M4 and SigA bound to the primary IgA-specific Fc receptor, Fc α RI (CD89). The structural data are complemented by SPR and mutagenesis data. The structures provide insight into mechanisms for streptococcal immune evasion as well as constraints on the ability of SigA to activate immune signaling via Fc α RI. Although data on IgA-receptor interactions has been known for quite a while with respect to monomeric (serum) IgA, it is exciting to begin to see structural data helps that helps to illuminate the biological activities of SigA, which plays a very important role in mucosal environments. Overall, the data is of high quality and the conclusions are mostly sound, although there are a few issues that need to be addressed and several areas of the text that need additional explanation or context.

Major points:

- All of the SPR data are shown as raw sensorgrams only without any fits or reported binding parameters (e.g., on- or off-rate constants or KD values). These data should certainly be fitted to the appropriate binding equations, with the fits shown overlaid on the raw data, and the relevant parameters reported with error ranges.
- Page 9, lines 159-160: the statement that there is a subset of residues in which one copy interacts with IgA and the other stabilizes the coiled-coil seems to suggest that M4 forms an asymmetric dimer—is that the case? The structure shown in Fig 2C seems to show a symmetric coiled-coil. If the SigA-binding region of M4 is indeed a symmetric dimer, please explain this more clearly and/or include a figure (perhaps in the supplementary information) that illustrates this phenomenon.
- In the discussion it would be of interest for the authors to mention other SigA-binding bacterial proteins involved in immune evasion, such as the pneumococcal protein SpsA and the streptococcal choline-binding protein A (from *S. pneumoniae*), each of which binds to the pIgR/SC directly rather than to the Fc α regions of SigA. It would be interesting to compare and contrast the different mechanisms of immune evasion and/or invasion mediated by these different types of SigA-binding bacterial proteins.
- The section of the discussion addressing the implications of the SigA:Fc α RI structure on immune signaling through Fc α RI needs a lot of work. It seems that the authors are suggesting on lines 294-295 that binding of two Fc α RI receptors to a single mIgA activates an inflammatory response from an Fc α RI-expressing immune cell, but this is incorrect. The C-termini of the two Fc α RI receptor chains bound to a single mIgA are much too far apart (~125 Å) to trigger a signaling response. Instead, the interaction of multiple IgAs bound to a multivalent target will position the Fc α regions close enough together that the bound Fc α RI receptors will form a cluster in the immune cell membrane. If the clustered Fc α RI molecules reach a sufficient density in the immune cell membrane, the associated

Fc γ R chains will be dislodged from the membrane, unmasking the phosphorylation site to allow signaling to be initiated. (The possibility mentioned in lines 304-306 is in fact the main way that mIgA activates Fc α RI signaling, and this might also occur with SIgA).

- I suspect the same issue would arise with SIgA; even though there are two distinct binding sites for Fc α RI per SIgA, these are distant from one another and would also be unable to trigger immune responses from a single Fc α RI-bound SIgA. It may be relevant to look at the likely orientation of the membrane-bound Fc α RI molecules in the context of the SIgA molecule to see whether it's likely that two membrane-embedded Fc α RI molecules could even simultaneously engage SIgA. It is also possible that the much bulkier SIgA is sterically precluded from approaching closely enough to other SIgA molecules to allow sufficiently dense Fc α RI clustering, which may be why SIgA is not good at triggering Fc α RI activation except in the context of a co-receptor such as Mac-1.
- It also seems highly likely from the modeling presented in Figure 6 that Fc α RI at the surface of an immune cell would be sterically prevented from engaging with M4-bound SIgA via the Fc α RI site unless M4 is extraordinarily flexible, which isn't clearly stated in the text. Unlike some related immune receptors (e.g., GPVI on platelets), Fc α RI does not have a long stalk region between the Ig-like domains and the transmembrane helix that might allow more flexibility to engage hard-to-access binding sites.
- In Table 1, why are the protein B-factors listed as 0 for both structures?
- Also in Table 1, why are bond lengths and bond angles for the 8SKV structure listed as 0?

Minor points:

- Page 3, line 33: the phrasing of this sentence is not very clear: "Whereas the binding of host Fc α Rs is likely to elicit a favorable immune response..."--instead of 'favorable' perhaps a more specific terms such as 'effector' or 'pro-inflammatory' response would be clearer.
- Page 5, lines 66-67: This sentence seems to indicate that tonsillitis results in 160,000 deaths per year; this should be rewritten to make it clear that invasive GAS may cause as many as 160,000 deaths per year worldwide.
- There are two locations that use the term 'Apr4' instead of M4—please change this nomenclature in the title of Figure 2 and on page 9, line 145.
- The caption of Figure 6 mentions that SC domain 2 is indicated by a circle and exposed near the outermost surface of the bacterium—what is the importance of this point? I did not see that SC domain 2 was mentioned at all in the text, so it has no context in this figure (perhaps this is related to the sentence on lines 288-290?).

Reviewer #3 (Remarks to the Author):

The manuscript by Liu and Stadtmueller describes cryo-EM structures of secretory IgA dimers (and secretory chain) in complex with the *Streptomyces pyogenes* cell wall-attached virulence factor M4 where they resolve a minimal IgA interacting peptide of M4. Interestingly, only a single M4 peptide binds dimeric IgA at a site which overlaps but is distinct from the CD89/FCalphaR1 binding site (known previously from X-ray studies of monomeric IgA in complex with two CD89's). The authors also describe a cryo-EM structure for two CD89 receptor ectodomains bound to the IgA dimer complex rather than two per monomer as observed by crystallography, which is due to steric hindrance of the J chain and SC. The complexes are informative for understanding how the virulence factor contributes to evasion of the immune system and shows that complexes with both CD89 and M4 may be possible. The authors also applied the knowledge of SIgA-M4 structure to the interactions between SIgA and other bacterial proteins such as M22 and SSL7. Overall, the work is of high-quality and of interest to the broad audience of Nature Communications.

Specific points:

1. The Introduction and Discussion, though providing important context, could potentially be shortened to better effect.
2. Line 145 and title of Fig. 2 "Apr4" should be "Arp4". Perhaps using the name 'M4' throughout the whole manuscript is better.
3. Line 294-296. "CD89 activity is thought to occur when multiple copies of the receptor bind to the antibody Fca and thus, it has been proposed that M4 occludes CD89 binding thereby blocking downstream effector functions." A literature reference will be useful here.
4. Map-model FSC curves and angular distribution plots should be provided for both structures as supplementary figures. The map-model FSC in Table is given at the 0.143 threshold, but the accepted threshold for map-model is 0.5.
5. The structure interpretations will be best supported by providing supplementary figures showing map quality for important interactions at interfaces for sIgA/M4 and sIgA/CD89.

REVIEWER COMMENTS

Reviewer #1 (Remarks to the Author):

Secretory IgA (SIgA) protects against infections at mucosal surfaces and has a more complex structure than monomeric IgA (mIgA), which is found in serum and has poorly understood function. For both forms of IgA, the human phagocyte receptor CD89 binds to the Fc part, an interaction that may promote phagocytosis *in vivo*. Moreover, both forms of IgA bind streptococcal IgA-Fc-binding surface proteins, which target a site that overlaps with the CD89-binding site. Special interest has been devoted to IgA-binding streptococcal M proteins, which belong to a family of coiled-coil proteins essential for virulence in the human pathogen *S. pyogenes*, a common cause of throat and skin infections. Because IgA-binding M proteins such as M4 inhibit the binding between mIgA and CD89, it has been suggested that these M proteins interfere with IgA-mediated phagocytosis. However, little is known about the *in vivo* function of IgA-binding M proteins.

While earlier studies were focused on mIgA, Liu and Stadtmueller studied the interaction of CD89 and M4 with SIgA, arguing that such studies are particularly relevant, because SIgA has a key role in preventing infections at mucosal surfaces. Cryo-EM of complexes convincingly demonstrated that the stoichiometry was 1:2 for the SIgA-CD89 complex, as expected from modeling, but for the SIgA-M4 complex it was unexpectedly found to be 1:1, implying that only one of the predicted M4 binding sites is accessible in SIgA. A likely explanation is that the J chain in SIgA sterically interferes with binding of M4, but not CD89, to the second region. This interesting result sheds new light on the binding properties of SIgA and may have implications for the poorly understood role of streptococcal IgA-binding proteins. Specifically, the data suggest that M4 may not interfere with phagocytosis promoted by interaction between SIgA and CD89. However, it is very uncertain whether SIgA plays a significant role in the defense against local *S. pyogenes* infections (see below), so the biological relevance of the findings is not clear. Overall, the intriguing molecular data reported in this ms would benefit from being considered in the context of *S. pyogenes* infections. In any case, it is of much interest that this ms reports the first structure of an M protein-IgA complex, with verification of structural data by site-specific mutagenesis.

Author response: We thank the reviewer for the positive comments and constructive suggestions. We have implemented suggestions in a revised manuscript that we feel better communicates the functional significance of our findings and better balances discussion of mIgA and SIgA interactions with *S. pyogenes*. Specific comments are addressed below.

Major comments:

1) The authors conclude that SIgA binds two molecules of CD89 but only one molecule of M4, implying that a complex M4-SIgA-CD89 can be formed even in the presence of an

excess of M4. This conclusion is based exclusively on comparisons of the two cryo-EM structures and should be substantiated by other means, e.g., by studying the binding properties of SIgA bound to immobilized M4 or by the characterization of soluble complexes.

Author response: Our determination of M4-SIgA and CD89-SIgA stoichiometry is based on cryoEM experiments and subsequent structural analysis. We feel this is a robust approach for determining stoichiometry for reasons listed below (1-2). As for the possibility that an M4-SIgA-CD89 complex can be formed; the existence of the M4-SIgA and CD89-SIgA structures and associated modeling indicate that it is sterically possible in solution; however, as detailed below (3-4), characterizing this complex in solution is challenging and we feel that modeling is sufficient for the level of interpretation presented in the text– the original and revised manuscripts state that sterically, M4-SIgA-CD89 complex formation appears feasible, but also acknowledge that it remains unknown if it does happen. The revised discussion lists variables that could contribute to the probability of a M4-SIgA-CD89 forming in vivo.

1. CryoEM is a commonly accepted solution-based approach for characterizing soluble proteins and complexes. Protein complexes are maintained in physiological buffer (e.g. pH=7.5 and salt = 150mM) and frozen in vitreous ice, which facilitates imaging and analysis of heterogeneous samples. Single particle analysis of 2 million particles resulted in 50 2D classes, 38 of which were used for 3D reconstruction. Careful inspection of our 2D classes and our resulting 3D map revealed no evidence for M4 protein bound at the F_{CD} binding site.
2. As described in the text, the JC, which binds each IgA uniquely, sterically clashes with M4 modeled at the F_{CD} binding site. Structurally, M4 could not bind the F_{CD} site unless the JC (in SIgA) was conformationally flexible and could move out of the way. This is unlikely because dimeric IgA is assembled in plasma cells through a chaperone assisted process in which the JC and IgA heavy chains fold together and form a disulfide-stabilized complex. Furthermore, the JC motif that occludes M4 binding adopts a beta hairpin structure. While residues in the turn form a loop, the hairpin's beta-sheet interactions, as well as their interactions with IgA are expected to limit the mobility of this region (along with electrostatic interactions between loop residues and the IgA heavy chain). In sum, multiple lines of structural evidence indicate that the JC is incompatible with M4 binding F_{CD} .
3. We have tried to characterize the SIgA complexes with M4, CD89 and M4/CD89 using size exclusion elution chromatography with in-line multi angle light scattering (SEC-MALS), which can provide the absolute molecular weight of proteins and complexes eluting from an SEC column and could, in theory, validate the observed M4-SIgA stoichiometry and demonstrate existence of a soluble M4-SIgA-CD89

complex. However, SEC elution profiles for SIgA-M4 and SIgA-CD89 (and SIgA-M4/CD89) do not produce tight monodisperse peaks. Rather, complexes elute as broad peaks containing mixtures of bound and unbound species. Some degree of complex dissociation on SEC columns is not uncommon since SEC is not an equilibrium experiment (buffer does not contain complex components); however, this limits our ability to unambiguously isolate a SIgA-M4/CD89 complex and, together with heterogeneous glycosylation, limits the accuracy of MALS analysis, making the experiment intractable.

4. Reasoning that CD89 could only bind SIgA that was bound to one M4, we attempted SPR experiments in which we immobilized soluble M4 on the surface on an SPR sensor and tested SIgA (and subsequently CD89) binding; this contrasts with experiments described in the manuscript, in which SIgA or mIgA are immobilized. Unfortunately to date, we have not been able to obtain high quality data for M4-SIgA binding due to technical limitations associated with experiments in which M4 is immobilized (e.g. non-specific interactions between SIgA and the sensor chip). In related experiments, SIgA was immobilized, M4 was injected and subsequently CD89 was co-injected while maintaining the same concentration of M4 (frame inject). While response was detected upon the addition of CD89 (not shown) it was not possible to be sure that CD89 was binding only to M4-SIgA complexes.

2) The authors stress the difference between the situation in blood, where *S. pyogenes* would encounter mIgA, and the situation on mucosal surfaces, where the bacteria would encounter SIgA. Since *S. pyogenes* only rarely enters blood, it may appear reasonable to focus on the interaction with SIgA. However, a local infection, such as a 'strep throat', will trigger an inflammatory response that results in massive exudation of plasma proteins and recruitment of neutrophils. Thus, the bacteria will encounter mIgA and phagocytes in the inflamed tissue, potentially allowing proteins such as M4 to interfere with mIgA-mediated phagocytosis. In contrast, it is uncertain whether SIgA plays a role in this situation. This argument becomes even stronger when it comes to skin infections (impetigo), since *S. pyogenes* will only encounter exudated mIgA in that case. Of note, impetigo is the most common *S. pyogenes* disease in many parts of the world and strains that bind IgA-Fc typically cause infections of the skin, not the throat, so strains that bind IgA may in most infections not encounter SIgA. It is also noteworthy that M4 binds to mIgA with an affinity that is more than 10-fold higher than for SIgA (ref. 26), suggesting that M proteins have evolved to selectively bind mIgA. The situation is even more striking for the *Streptococcus agalactiae* (GBS) protein beta (Bac), which has high affinity for mIgA but binds SIgA only weakly. While these arguments focus interest on the interaction between pathogenic streptococci and mIgA, they do not exclude that the features of SIgA described by Liu and Stadtmueller play a role in certain situations, although the available evidence indicates that mIgA is biologically more relevant in *S. pyogenes* infections.

Author Response: We thank the reviewer for comments that highlight the importance of thoroughly considering mIgA interactions with GAS when interpreting and communicating our data. Accordingly, we have revised the Introduction and Discussion sections to provide more rigorous and balanced presentation of when, where and how, mIgA and SIgA interactions GAS might occur in the host, as well as potential outcomes of those interactions. We feel this more balanced approach, including discussion of potential differences between mIgA and SIgA-GAS interactions, highlights the relevance of our data and of studying SIgA-GAS interactions (see further comments below).

1. In our revised Introduction have clarified when and where monomeric (m) IgA and/or SIgA could be encountered by GAS and have specifically mentioned the presence of mIgA in inflamed tissues and GAS skin infections (e.g. Impetigo). However, we have not found clear evidence in the literature suggesting that skin infections have provided the dominant evolutionary pressure for IgA interactions with GAS, or that their prevalence excludes the possibility that SIgA is relevant in the mucosal infections. Specifically, our review of recent literature suggests, emm4 is significantly correlated with noninvasive GAS infections of respiratory origin (Rafei et al, Infection, Genetics and Evolution, 2022). For example, Emm4 was the most frequent genotype associated with pharyngitis in Korea (Kim et al, J med microbio, 2019); emm4 was the most frequently found emm type for isolates of respiratory tract origin from a study conducted in Austria in 2003 (Eisner, Diagnostic Microbiology and Infectious Disease, 2006); emm4 was more prevalent in non-invasive isolate (17.4%) than invasive isolates (compared to 4.2% invasive) in Ireland (Meehan et al, J Infect, 2018); a 7-year GAS study in Spain reported pharyngitis to be the main clinical presentation and emm4 was the most frequent one associated with pharyngitis (Tamayo et al., J Infect, 2014); emm4 contributed to 23.5% of pharyngotonsillitis in Japan, compared to 2.7% of invasive disease (Wajima et al., J med microbio, 2008). While these reports do not prove whether mIgA and/or SIgA (or non-mucosal versus mucosal infections) is a major evolutionary driver of host-GAS interactions, we think they support the notion that we simply do not know the relative contribution of each and therefore data on both are relevant.
2. Our revised introduction also mentions the prevalence of *S. pyogenes* asymptomatic carriers. According to the analysis conducted by Shaikh et al. (Pediatrics, 2010), it was found that the prevalence of *S. pyogenes* carriage (tested by throat culture) among asymptomatic, healthy children without any signs or symptoms of pharyngitis was 12% (95% CI 9-14%). In these cases, GAS may encounter SIgA but might not trigger inflammatory responses that promote interactions with mIgA.

3. The reviewer brings up the possibility that mIgA interactions with M4 have evolved to be stronger than interactions with SIgA, and references data that M4 binds mIgA with 10-fold higher affinity than SIgA. Structurally, we do not find evidence to suggest that M4 would bind different residues on mIgA versus SIgA, and we note that other publications focusing on related protein, M22, found similar binding affinity for mIgA and SIgA (Sandin et al., *J Immunol*, 2002). Nonetheless, without a structure of mIgA bound to M4, we cannot completely rule out the possibility that there are differences. Given this, we have tested soluble M4 binding to mIgA and SIgA using SPR and found KDs to be very similar (~15nM and 20nM, respectively; revised Supplementary Fig. S5). Although there is a 5nM difference in KD, this is minimal compared to the ~ 130nM change KD observed for the individual M4 point mutants. To us, these similarities suggest that mIgA and SIgA contain molecularly equivalent M4 binding sites and that enhanced binding affinity specifically for mIgA was not selected for. However, we recognize that the two binding sites on mIgA might promote higher avidity interactions with GAS than the single site on SIgA. While it remains to be tested IF two copies of M4 on a bacterium surface can bind one IgA, higher avidity may provide an explanation for published reports indicating that the affinity of mIgA is 10-fold higher than SIgA. The binding of both sites on a single mIgA would also block all CD89 binding sites, consistent with published data suggesting that inhibiting CD89-dependnet effector functions is at least one of M4's functions. We have presented this model in our revised discussion.
4. Finally, the reviewer acknowledges that published data focusing on mIgA interactions with *S. pyogenes* do not exclude the possibility that "the features of SIgA described by Liu and Stadtmueller play a role in certain situations although the available evidence indicates that mIgA is biologically more relevant in *S. pyogenes* infections." We agree that plenty of data suggests GAS-SIgA interactions are biologically relevant; however, we also feel that "available evidence" is limited, making it difficult to judge whether mIgA or SIgA is "more" biologically relevant. Indeed, it seems that both must be considered.

Previous publications have shown the quantity of SIgA remaining bound to *S. pyogenes* cultures is equivalent to that of mIgA (e.g. Lindahl & Akerstrom, *Molecular Microbiology*, 1989), which together with our findings and high concentrations of SIgA in the human upper respiratory tract, imply that *S. pyogenes* located in mucosal secretions will encounter and bind SIgA. Our data further indicate that these interactions are constrained in a manner that is likely to orient all copies of bound SIgA in a similar orientation. To the best of our knowledge this is not something that has been described for SIgA interactions with any microbe, and as noted in the discussion, has implications for both the

host and the microbe. For example, even if SIgA plays a minor role compared to mIgA in some aspects of GAS infection or evolution (though as noted, available evidence supporting this is limited) SIgA may play a major role in another aspect of infection or evolution, perhaps providing advantages for the microbe in the mucosa that mIgA cannot provide. Our work provides structural rationale supporting this as well as the foundation for exploring these and other possibilities.

Other comments:

3) Discussion: It is very long, so the authors may consider removing the SSL7 text (lines 242-257), which is interesting but does not fit well into the paper. -- Because the interaction of M4 with SIgA may be of limited importance, as compared with the binding to mIgA (see above), it becomes problematic that the Discussion puts such strong emphasis on this interaction. -- The argument on lines 315-319 is unclear. -- Concerning the role of IgA (mIgA or SIgA) bound to an M protein, the authors may consider the possibility that it is unrelated to the interaction between IgA and CD89.

Author Response: Thank you. We have shortened discussion on SSL7 and moved it to the end of the Discussion section so that more relevant topics are presented first. We did not eliminate discussion because we feel that comparisons to IgA-SSL7 complexes are structurally (and evolutionarily) relevant. M4, SSL7 and CD89 are structurally and functionally diverse proteins that each engage a similar site on IgA and thus from the perspective of IgA binding to microbes, antigens and host cells, we feel it would be incomplete to eliminate discussion on SSL7, especially because presented modeling suggests that the stoichiometry of SSL7 interactions with SIgA would differ from M4- SIgA interactions. Furthermore, another reviewer has asked us to expand the discussion of SIgA-binding proteins and to include *S. pneumoniae* CbpA interactions with SIgA (see below); in the revised manuscript, we have combined discussion on SSL7 and CbpA into one paragraph.

As noted above, we have tried to balance discussion on mIgA and SIgA interactions with M4 in the revised manuscript and clarify when and where interactions with SIgA would be relevant. A significant finding of our work is that GAS will engage SIgA differently than mIgA; this is a new observation and thus we feel that discussion of possible outcomes is not problematic, but is necessary, to communicate the significance of the structure to a broader audience. Additionally, we have reworded the Discussion Section to include a paragraph on CD89-SIgA interactions independent from interactions with M4. We agree that it is possible that the M4-SIgA interactions are unrelated to CD89 function and our original and revised manuscripts discuss numerous other potential functions (adhesion, sequestration of Fabs, etc.). Still, we cannot rule out the possibility that M4-SIgA binding influences CD89

binding and/or activity; thus, we proposed this as a possibility. We hope that more careful wording in the revised manuscript resolves this issue.

4) The numbering of aa residues in the IgA-binding regions of M4 and M22 is permuted one step, as compared to the correct one (see refs. 6 and 5). This must be corrected throughout, in text and figures.

Author Response: Thank you for pointing this out; we have corrected this error throughout the text and in the PDB file.

5) line 35: "...interactions that allow...". A more cautious phrasing would be "...interactions that may allow...".

Author Response: Thank you for pointing this out; we have implemented this change.

6) lines 66-68: It is correct that *S. pyogenes* tonsillitis is much more common than invasive infections, but it is invasive infections, not tonsillitis, that cause over 160,000 deaths per year.

Author Response: Thank you for pointing this out; we have re-written this part of the introduction.

7) line 84: the region at the N-terminus of an M protein is not the most immunogenic region but the key site for protective antibodies (Lannergård et al., Cell Host Microbe 2011).

Author Response: Thank you; we have corrected this.

8) lines 88-89: the words 'including five C-terminal residues of the HVR' are unclear and can be removed.

Author Response: We have removed this based on literature indicating that the definition of HVR varies between different m proteins.

9) line 105: '...in contrast to previously proposed...'. Already in ref. 24, it was reported that the (overlapping) sites interacting with the CD89 and M4 are nonidentical.

Author Response: Thank you for pointing this out, it was our intention to emphasize differences between the specific residues in the binding site that we mapped in our structures and what was previously proposed in domain-swap experiments, which identified multi-residue motifs. We have removed "in contrast to previously proposed...".

10) line 236: 'prior mutational analysis'. To my knowledge, such analysis has not been reported. Thus, the word 'mutational' can be deleted.

Author Response: By "mutational analysis" we meant the deletion derivatives/mutants that were made to determine the necessary IgA-binding region in M4 in Johansson et al, *J. Immunol*, 1994. However, we have removed the word "mutational" as suggested.

11) line 267: the words 'outer membrane' should be removed. Since *S. pyogenes* is a Gram-positive bacterium, it does not have an outer membrane.

Author Response: Thank you; we have removed "outer membrane."

12) line 281: "...surface antigens, such as..." Most of the examples given are not surface proteins, but proteins released from the bacteria.

Author Response: We have clarified our wording and deleted exotoxin and SLO, which are secreted proteins. SFb, SpyCEP and SCP are surface associated, although SCP can also be secreted in some strains.

13) line 351: Like other M proteins, M4 has a signal peptide but does not have a 'propeptide'

Author Response: Thank you. We initially had used the term "Propeptide" because the Uniprot database (P13050) annotates this region as "Propeptide." We have changed it to "cell wall sorting signal" in the text.

▶ Signal		1-41	
▶ Chain	PRO_0000005591	42-356	IgA receptor
▶ Modified residue		356	Pentaglycyl murein peptidoglycan amidated threonine  PROSITE-ProRule Annotation
▶ Propeptide	PRO_0000005592	357-386	Removed by sortase  PROSITE-ProRule Annotation

Reviewer #2 (Remarks to the Author):

This manuscript describes separate cryoEM structures of SIgA bound to a streptococcal immune evasion protein M4 and SIgA bound to the primary IgA-specific Fc receptor, Fc α RI (CD89). The structural data are complemented by SPR and mutagenesis data. The structures provide insight into mechanisms for streptococcal immune evasion as well as constraints on the ability of SIgA to activate immune signaling via Fc α RI. Although data on IgA-receptor interactions has been known for quite a while with respect to monomeric (serum) IgA, it is exciting to begin to see structural data helps that helps to illuminate the biological activities of SIgA, which plays a very important role in mucosal environments. Overall, the data is of high quality and the conclusions are mostly sound, although there are a few issues that need to be addressed and several areas of the text that need additional explanation or context.

Author Response: We thank the review for positive feedback and have implemented suggested changes as noted below.

Major points:

- All of the SPR data are shown as raw sensorgrams only without any fits or reported binding parameters (e.g., on- or off-rate constants or KD values). These data should certainly be fitted to the appropriate binding equations, with the fits shown overlaid on the raw data, and the relevant parameters reported with error ranges.

Author Response: Thank you; we have fit SPR data with a 1-1 binding model, which is shown along with errors in Supplementary Fig. S3 and have modified the methods accordingly.

- Page 9, lines 159-160: the statement that there is a subset of residues in which one copy interacts with IgA and the other stabilizes the coiled-coil seems to suggest that M4 forms an asymmetric dimer—is that the case? The structure shown in Fig 2C seems to show a symmetric coiled-coil. If the SigA-binding region of M4 is indeed a symmetric dimer, please explain this more clearly and/or include a figure (perhaps in the supplementary information) that illustrates this phenomenon.

Author Response: M4 adopts a symmetric coiled-coil. Several residues, including Q66, stabilize the coiled-coil and interactions with the IgA. We have clarified this in the revised text. For example, Q66 on helix-b of M4: OE1 is interacting with S356 on IgA, while NE2 is forming hydrogen bond with Y65 on helix-a of M4.

- In the discussion it would be of interest for the authors to mention other SIgA-binding bacterial proteins involved in immune evasion, such as the pneumococcal protein SpsA and the streptococcal choline-binding protein A (from *S. pneumoniae*), each of which binds to the pIgR/SC directly rather than to the Fc α regions of SIgA. It would be interesting to

compare and contrast the different mechanisms of immune evasion and/or invasion mediated by these different types of SIgA-binding bacterial proteins.

Author Response: We thank the reviewer for this suggestion and have added two sentences related to SpsA in the discussion; as the reviewer points out, interactions with pIgR and SC support the notion that bacterial interactions with SIgA are biologically significant and functionally diverse. However, owing to other reviewers' suggestions that the discussion be shortened (even removing discussion on SSL7) we have not greatly expanded upon this point.

- The section of the discussion addressing the implications of the SIgA:Fc α RI structure on immune signaling through Fc α RI needs a lot of work. It seems that the authors are suggesting on lines 294-295 that binding of two Fc α RI receptors to a single mIgA activates an inflammatory response from an Fc α RI-expressing immune cell, but this is incorrect. The C-termini of the two Fc α RI receptor chains bound to a single mIgA are much too far apart (~125 Å) to trigger a signaling response. Instead, the interaction of multiple IgAs bound to a multivalent target will position the Fc α regions close enough together that the bound Fc α RI receptors will form a cluster in the immune cell membrane. If the clustered Fc α RI molecules reach a sufficient density in the immune cell membrane, the associated Fc γ chains will be dislodged from the membrane, unmasking the phosphorylation site to allow signaling to be initiated. (The possibility mentioned in lines 304-306 is in fact the main way that mIgA activates Fc α RI signaling, and this might also occur with SIgA).

Author Response: Thank you for pointing this out. Our original comments were intended to balance possibilities if free IgA versus antigen-bound IgA were to bind CD89(s), but as written this was certainly not clear. Indeed, the reviewer is correct that the modeled distance between CD89 signaling domains upon binding mIgA is likely too far to initiate signaling (for mIgA). We have addressed this in the revised Introduction by adding a paragraph that summarizes models for CD89/ Fc α RI signaling and have further discussed potential differences between SIgA and mIgA interactions with CD89 in the revised Discussion section. Additionally, we have attempted to better balance discussion of CD89 versus M4 throughout the text.

- I suspect the same issue would arise with SIgA; even though there are two distinct binding sites for Fc α RI per SIgA, these are distant from one another and would also be unable to trigger immune responses from a single Fc α RI-bound SIgA. It may be relevant to look at the likely orientation of the membrane-bound Fc α RI molecules in the context of the SIgA molecule to see whether it's likely that two membrane-embedded Fc α RI molecules could even simultaneously engage SIgA. It is also possible that the much bulkier SIgA is sterically precluded from approaching closely enough to other SIgA molecules to allow sufficiently dense Fc α RI clustering, which may be why SIgA is not good at triggering Fc α RI activation except in the context of a co-receptor such as Mac-1.

Author Response: Thank you for pointing this out. We have measured the distance between the C-terminal residues of the CD89 D2 domain in our CD89-SIgA structure and reported the value (108Å) in the revised manuscript. This value is slightly larger than the 99Å distance separating CD89 C-termini in a modeled a CD89-SIgA complex (Wang et al. *Cell Research*, 2020). Even though this distance is shorter than the 124 Å reported for CD89-mIgA complexes, it is unclear if this is the case in vivo and/or if this distance would readily promote signaling. We have noted this in the revised discussion.

As described above, we have clarified that pro-inflammatory activation of CD89 is almost certainly dependent on binding to IgA-antigen complexes. Our CD89-SIgA structure may facilitate improved modeling experiments in the context of a cell membrane. We have considered these experiments; however, we have concluded they would be computationally expensive, beyond our expertise and beyond the scope of this paper. In lieu of those data, we have clarified differences between mIgA and SIgA binding to CD89 in the revised Results section and discussed how that could influence signaling in the revised Discussion section. Although the bulkiness of SIgA could limit SIgA-antigen interactions with CD89, we note that previous modeling demonstrates that the bent and tilted geometry of SIgA will direct antigen interactions away from the CD89 binding sites; thus, it also appears plausible that evolution has preserved CD89 access to SIgA. It also seems likely that epitope spacing and the relative orientation of bound IgA could be different for each antigen, and thus impact CD89 binding. Our revised discussion includes a paragraph devoted to CD89-IgA interactions that discusses some of these possibilities.

- It also seems highly likely from the modeling presented in Figure 6 that Fc α RI at the surface of an immune cell would be sterically prevented from engaging with M4-bound SIgA via the Fc_{cd} site unless M4 is extraordinarily flexible, which isn't clearly stated in the text. Unlike some related immune receptors (e.g., GPVI on platelets), Fc α RI does not have a long stalk region between the Ig-like domains and the transmembrane helix that might allow more flexibility to engage hard-to-access binding sites.

Author Response: We are unaware of data describing the flexibility of M4 on the GAS surface and thus, it remains a variable in this model (as noted in the original and revised texts). We agree that there is a high chance that the Fc_{CD} site is sterically occluded; however, we felt it was necessary to mention the possibility since there are other unknown variables. In the revised text we have attempted to better balance discussion on the possible outcomes of SIgA-M4 binding.

- In Table 1, why are the protein B-factors listed as 0 for both structures?

Author Response: Thank you; we have corrected this error.

- Also in Table 1, why are bond lengths and bond angles for the 8SKV structure listed as 0?

Author Response: Thank you; we have corrected this error.

Minor points:

- Page 3, line 33: the phrasing of this sentence is not very clear: "Whereas the binding of host Fc α R_s is likely to elicit a favorable immune response..."--instead of 'favorable' perhaps a more specific terms such as 'effector' or 'pro-inflammatory' response would be clearer.

Author Response: Thank you; we have changed it to "effector".

- Page 5, lines 66-67: This sentence seems to indicate that tonsillitis results in 160,000 deaths per year; this should be rewritten to make it clear that invasive GAS may cause as many as 160,000 deaths per year worldwide.

Author Response: Thank you; we have corrected this error.

- There are two locations that use the term 'Apr4' instead of M4—please change this nomenclature in the title of Figure 2 and on page 9, line 145.

Author Response: Thank you; we have corrected this error.

- The caption of Figure 6 mentions that SC domain 2 is indicated by a circle and exposed near the outermost surface of the bacterium—what is the importance of this point? I did not see that SC domain 2 was mentioned at all in the text, so it has no context in this figure (perhaps this is related to the sentence on lines 288-290?).

Author Response: Thank you for pointing this out. Yes, it's related to line 288-290 in the original manuscript; we also added more context for SC D2 domain in the revised discussion.

Reviewer #3 (Remarks to the Author):

The manuscript by Liu and Stadtmueller describes cryo-EM structures of secretory IgA dimers (and secretory chain) in complex with the *Streptomyces pyogenes* cell wall-attached virulence factor M4 where they resolve a minimal IgA interacting peptide of M4. Interestingly, only a single M4 peptide binds dimeric IgA at a site which overlaps but is distinct from the CD89/FC α R1 binding site (known previously from X-ray studies of monomeric IgA in complex with two CD89's). The authors also describe a cryo-EM structure for two CD89 receptor ectodomains bound to the IgA dimer complex rather than two per monomer as observed by crystallography, which is due to steric hindrance of the J chain and SC. The complexes are informative for understanding how the virulence factor contributes to evasion of the immune system and shows that complexes with both CD89 and M4 may be possible. The authors also applied the knowledge of SIgA-M4 structure to the interactions between SIgA and other bacterial proteins such as M22 and SSL7. Overall, the work is of high-quality and of interest to the broad audience of Nature Communications.

Author Response: We thank the review for positive feedback and have implemented suggested changes as noted below.

Specific points:

1. The Introduction and Discussion, though providing important context, could potentially be shortened to better effect.

Author Response: Thank you; we have reworded the Introduction and Discussion sections. In doing so, we had to balance this request with another reviewers' request to add discussion on other IgA binding proteins and to clarify potential functional outcomes of SIgA interactions with CD89 and M4; thus, while neither section is substantially shorter than the original manuscript, we have shortened some paragraphs and feel that overall, the writing is more concise and the content is improved.

2. Line 145 and title of Fig. 2 "Apr4" should be "Arp4". Perhaps using the name 'M4' throughout the whole manuscript is better.

Author Response: Thank you; we have corrected this error.

3. Line 294-296. "CD89 activity is thought to occur when multiple copies of the receptor bind to the antibody Fc α and thus, it has been proposed that M4 occludes CD89 binding thereby blocking downstream effector functions." A literature reference will be useful here.

Author Response: Thank you; we have added a reference.

4. Map-model FSC curves and angular distribution plots should be provided for both structures as supplementary figures. The map-model FSC in Table is given at the 0.143 threshold, but the accepted threshold for map-model is 0.5.

Author Response: Thank you; we have corrected this error.

5. The structure interpretations will be best supported by providing supplementary figures showing map quality for important interactions at interfaces for sIgA/M4 and sIgA/CD89.

Author Response: Thank you; we have expanded upon previous figures that showed map quality for representative residues at the interface (e.g. M4 K68 and SIgA A360; CD89 Y35, F56 and SIgA F443).

REVIEWERS' COMMENTS

Reviewer #1 (Remarks to the Author):

The revised ms is fine. The conclusion, that only one binding site for M4 is accessible in SIgA, should ideally have been confirmed by a second method, but I appreciate that such analysis is not possible for the time being. As for the relative biological role of mIgA and SIgA in interactions with IgA-binding M proteins, the new arguments of the authors are thoughtful and emphasize that both interactions must be considered.

My only (minor) comments concern the new texts in the Discussion:

Lines 253-272: The authors describe new modeling data and SPR analysis, shown in Fig. S5. It is valuable to include a model showing that mIgA most likely binds two molecules of M4, but the accompanying text is too specialized, especially for a Discussion, so it would be valuable if the authors could shorten and/or simplify it.

Lines 317 ff: The authors mention that GAS strains with the IgA-binding M4 and M22 proteins are common, focusing interest on IgA-binding M proteins. This argument is strongly supported by data showing that not only M4 and M22 but five other purified M proteins (M11, M28, M48, M60, M85) bind IgA, and by the presence of a putative IgA-binding motif in several other M proteins (Stenberg et al., *Molec Microbiol.* 6, 1185; 1992; Sanderson-Smith et al., *J Infect Dis* 210, 1325; 2014).

Reviewer #2 (Remarks to the Author):

The authors have done a good job of addressing my concerns (and balancing the sometimes divergent concerns of all the reviewers). In particular, the portions of the manuscript discussing immune activation via FcαRI are more carefully written and read much better. Furthermore, the inclusion of the fits for the SPR data in Figure S3 is important, and the new SPR data on M4 binding to mIgA versus SIgA is a nice addition. The only minor point regarding SPR would be that it would be nice to include the details of the on-rate and off-rates determined from the kinetic analyses (in addition to the calculated KD values) in one or more tables, in order to allow comparison with other published SPR studies on IgA, but I do not feel that this is a requirement for publication.

In summary, this revision is a well written, impactful manuscript that will be of broad interest to a wide audience.

REVIEWERS' COMMENTS

Reviewer #1 (Remarks to the Author):

The revised ms is fine. The conclusion, that only one binding site for M4 is accessible in SIgA, should ideally have been confirmed by a second method, but I appreciate that such analysis is not possible for the time being. As for the relative biological role of mIgA and SIgA in interactions with IgA-binding M proteins, the new arguments of the authors are thoughtful and emphasize that both interactions must be considered.

My only (minor) comments concern the new texts in the Discussion:

Lines 253-272: The authors describe new modeling data and SPR analysis, shown in Fig. S5. It is valuable to include a model showing that mIgA most likely binds two molecules of M4, but the accompanying text is too specialized, especially for a Discussion, so it would be valuable if the authors could shorten and/or simplify it.

Author Response: Thank you, we have simplified the paragraph and modified the figure S5 legend accordingly.

Lines 317 ff: The authors mention that GAS strains with the IgA-binding M4 and M22 proteins are common, focusing interest on IgA-binding M proteins. This argument is strongly supported by data showing that not only M4 and M22 but five other purified M proteins (M11, M28, M48, M60, M85) bind IgA, and by the presence of a putative IgA-binding motif in several other M proteins (Stenberg et al., Molec Microbiol. 6, 1185; 1992; Sanderson-Smith et al., J Infect Dis 210, 1325; 2014).

Author Response: Thank you for pointing it out; we have mentioned the existence of other IgA-binding M proteins in the revised Introduction and revised discussion and included the two suggested references.

Reviewer #2 (Remarks to the Author):

The authors have done a good job of addressing my concerns (and balancing the sometimes divergent concerns of all the reviewers). In particular, the portions of the manuscript discussing immune activation via FcαRI are more carefully written and read much better. Furthermore, the inclusion of the fits for the SPR data in Figure S3 is important, and the new SPR data on M4 binding to mIgA versus SIgA is a nice addition. The only minor point regarding SPR would be that it would be nice to include the details of the on-rate and off-rates determined from the kinetic analyses (in addition to the calculated KD values) in one or more tables, in order to allow comparison with other published SPR studies on IgA, but I do not feel that this is a requirement for publication.

In summary, this revision is a well written, impactful manuscript that will be of broad interest to a wide audience.

Author Response: Thank you for the positive comments and suggestions. We have included the on-rate and off-rates in addition to the KD values for all SPR experiments in Table S2.